# Quantitative modeling of the effect of antigen dosage on B-cell affinity distributions in maturating germinal centers

Marco Molari[1], Klaus Eyer[2], Jean Baudry[3], Simona Cocco[1†], Rémi Monasson[1†]*

[1]Laboratoire de Physique de l'École Normale Supérieure, ENS, PSL University, CNRS UMR8023, Sorbonne Université, Université Paris-Diderot, Sorbonne Paris Cité, Paris, France; [2]Laboratory for Functional Immune Repertoire Analysis, Institute of Pharmaceutical Sciences, ETH Zurich, Zurich, Switzerland; [3]Laboratoire Colloides et Materiaux Divises (LCMD), Chemistry, Biology and Innovation (CBI), ESPCI, PSL Research and CNRS, Paris, France

**Abstract** Affinity maturation is a complex dynamical process allowing the immune system to generate antibodies capable of recognizing antigens. We introduce a model for the evolution of the distribution of affinities across the antibody population in germinal centers. The model is amenable to detailed mathematical analysis and gives insight on the mechanisms through which antigen availability controls the rate of maturation and the expansion of the antibody population. It is also capable, upon maximum-likelihood inference of the parameters, to reproduce accurately the distributions of affinities of IgG-secreting cells we measure in mice immunized against Tetanus Toxoid under largely varying conditions (antigen dosage, delay between injections). Both model and experiments show that the average population affinity depends non-monotonically on the antigen dosage. We show that combining quantitative modeling and statistical inference is a concrete way to investigate biological processes underlying affinity maturation (such as selection permissiveness), hardly accessible through measurements.

**\*For correspondence:**
monasson@lpt.ens.fr

[†]These authors contributed equally to this work

**Competing interests:** The authors declare that no competing interests exist.

## Introduction

Vaccines are undoubtedly one of the most effective preventive procedure ever developed and have even been used to eradicate diseases (*Greenwood, 2014*; *Nanni et al., 2017*). In many cases, vaccine-mediated protection can be directly linked to the generation of an antigen-specific antibody repertoire (*Nakaya et al., 2015*; *Li et al., 2014*), such as for tetanus toxoid (TT) vaccination (*WHO, 2017*; *Ershler et al., 1982*). The repertoire, a term detailing the present antibody variants within an organism, is adapted upon vaccination to include vaccine-specific clones (*Lavinder et al., 2014*; *Lee et al., 2017*). The processes that shape and expand this repertoire upon vaccination are highly complex and dynamic and are strongly linked to affinity maturation (AM) (*Keck et al., 2016*; *Tas et al., 2016*; *Allen et al., 2007a*). AM entails a series of mechanisms through which the immune system is able to produce potent high-affinity and antigen-specific antibodies (Abs) (*Victora and Nussenzweig, 2012*; *De Silva and Klein, 2015*; *Bannard and Cyster, 2017*; *Mesin et al., 2016*; *Eisen, 2014*; *Victora and Mesin, 2014*). Briefly speaking, AM is achieved through the combination of random mutations and selection for Antigen (Ag) binding. AM takes place in microanatomical structures, known as germinal centers (GCs). GCs are initially seeded by B-lymphocytes from the naive repertoire with sufficient affinity to bind the Ag. This initial affinity is achieved thanks to the great diversity of the immune repertoire, generated by processes such as VDJ recombination

(*Elhanati et al., 2015*). B-cells in GCs iteratively migrate through two areas, called the GC light and dark zones (LZ/DZ). In DZ, cells duplicate and are subject to a high mutation rate through a process known as *Somatic Hypermutation* (SHM). Cells then migrate out of DZ to LZ, where they are selected for Ag binding through a process involving interaction with follicular T-helper cells. Selected cells migrate then back to DZ for further duplications. This combination of random mutations and selection for Ag binding constitute a Darwinian evolutionary process, which progressively enhances the affinity of the B-cell population for the Ag.

In practice, AM is induced through administration of some dose of attenuated Ag, often mixed with adjuvants and other additives that have both immune-stimulatory effect and facilitate retention of Ag for longer periods of time (*Asensio et al., 2019*; *HogenEsch et al., 2018*; *Awate et al., 2013*; *Coffman et al., 2010*). Whilst the adjuvant and additives define the nature of the immune response (*Coffman et al., 2010*), Ag dose is a major variable in AM (*Eisen, 2014*; *Foote and Eisen, 1995*; *Kang et al., 2015*). High-affinity cells are discriminated and selected based on their capacity to bind Ag, and the amount of available Ag therefore tunes the strength of the applied Darwinian selection, that is defining the selection pressure (*Kang et al., 2015*; *Baer et al., 1954*; *Tam et al., 2016*). For example in reference (*Kang et al., 2015*), based on measurements of Abs affinity in rabbit sera following hapten immunization (*Eisen and Siskind, 1964*), the authors observed that average affinity decreased and heterogeneity increased with Ag dosage, suggesting that the latter was controlling the strength of selection: low and high dosages corresponded to, respectively, strong and weak selections (*Goidl et al., 1968*; *Nussenzweig and Benacerraf, 1967*; *Tam et al., 2016*). However, experimental evidence exists suggesting that Ag dosage has also a non-trivial effect on the efficacy of affinity maturation. This selection will be applied in the highly complex and dynamic environment of the immune response and the dose-response curve for some vaccines is not a saturating function of the Ag dose (*Rhodes et al., 2019*). Experiments showed that there was an intermediate range of concentrations for optimal stimulation of the immune system, leading the authors to advocate the development of data-informed models to guide the vaccine dose decision-making process, for example in the cases of tuberculosis, malaria, HIV (*Rhodes et al., 2019*). Models for AM were proposed to investigate this aspect and to help developing protocols in the field of vaccine design. Examples include the study of optimal immunization strategies against highly mutable pathogens such as HIV (*Shaffer et al., 2016*; *Wang, 2017*; *Wang et al., 2015*) and the influence of Ag administration kinetic on the humoral response (*Tam et al., 2016*); a review of Germinal Center Reaction models and their ingredients can be found in *Buchauer and Wardemann, 2019*.

A second open issue concerning AM is to characterize in a quantitative way the selection acting in the GC, in particular how *permissive* it is (*Bannard and Cyster, 2017*; *Mesin et al., 2016*; *Victora and Mouquet, 2018*; *Inoue et al., 2018*). Through mechanisms such as bystander activation (*Bernasconi, 2002*; *Eyer et al., 2020*; *Eyer et al., 2017*) GC selection can indeed allow intermediate- and low-affinity clones to survive (*Tas et al., 2016*). These phenomena generate a wider diversity than previously appreciated, especially when considering complex Ags displaying different epitopes (*Kuraoka et al., 2016*). In *Finney et al., 2018* for example the authors try to characterize the GC response to complex Ags such as influenza vaccine, as opposed to simple ones such as haptens. While in the latter case, a strong homogenizing selection and affinity maturation is observed, for complex Ags response is more polyclonal and a consistent part of the GC population (20–30%) is composed of low-affinity clones. This suggests a more permissive nature of the GC selection, in which even low-affinity clones have a non-zero probability of passing the selection. Permissiveness could for example be useful against mutable pathogens, where maintaining a pool of general cross-reactive cells might be a better strategy than only selecting for the best strain-specific binders.

In this paper, we tackle the question of how the Ag dosage and the time delay between subsequent vaccine injections can influence the quality of immunization, measured as the Ag affinity of the B-cell population that respond to a further antigenic challenge. Thanks to the technique developed in *Eyer et al., 2017* we were able to access full experimental affinity *distribution* of splenic Ab-secreting cells (Ab-SCs) extracted from mice following TT immunization. These distributions constitute a much more detailed information than other affinity measurement, such as average serum affinity, which only summarize them in a single number that is often related to their average. We introduce a computational model, inspired by previous work (*Wang et al., 2015*), that is capable of reproducing these distribution under different immunization schemes, in which both the Ag dosage and the delay between injections can be varied. We aim at studying the mechanisms underlying the

observed optimality of Ab affinity at intermediate dosages through detailed mathematical analysis of the model. In addition to this, our aim is to probe how restrictive GC selection is in our particular immunization protocol, and therefore we include in the model some parameters that encode for permissiveness and stochasticity. We use inference techniques to find the most likely value of the parameters given the observed data. This allow us to have information on quantities that are not directly measurable in experiments.

## Results

### Stochastic model for affinity maturation

We model the stochastic evolution of the distribution of binding energies of a population of B-cells during the affinity maturation (AM) process. A virtual population of B-cells in the GC is subject to iterative rounds of duplication, mutation and selection, see *Figure 1* (*Wang et al., 2015*). Each B-cell in our model is characterized by the binding energy $\epsilon$ between its receptor and the Ag; $\epsilon$ is measured in units of $k_B T$, where $k_B$ is Boltzmann constant and $T$ the organism temperature (This choice of unit is standard in biophysics, and allows one to simply express Boltzmann factors as $e^{-\epsilon}$; in practice, $1\,k_B T \simeq 10^{-24}$ kcal). This energy is related to the dissociation constant $K_d$ between the B-cell receptor and the Ag through $\epsilon = \log K_d$ is here expressed in Molar units, other choices of units would shift energies by a constant amount. Hence, lower energies correspond to higher affinities (For example, the dissociation constant $K_d = 1$ µM corresponds to the energy $\epsilon = -13.8$, and a tenfold decrease in affinity (with $K_d$ varying from 1 µM to 10 µM) corresponds to an increase in binding energy of 2.3). The main objective of our model is to track the evolution of the distribution of binding energies across the B-cell population, $\rho(\epsilon)$, during the GC maturation process. Tracking the full distribution is important for later comparison with experimental data, which themselves consist of affinity distributions. We now describe the main ingredients of the model.

### Ag dynamics

In the course of AM, the concentration $C$ of Ag varies over time, due both to gradual release from the adjuvant matrix and to decay and consumption (*Figure 2A*). At time of injection Ag molecules are trapped in the adjuvant matrix, which constitutes an Ag reservoir. Ag is then quickly released at a fast rate $k^+$. Due to recycling of Ag from surface of follicular dendritic cells (FDCs) to endosomal compartments (*Heesters et al., 2013*; *Mesin et al., 2016*) available Ag decays at a slow rate $k_{\emptyset}^-$, and are consumed by B-cells at a faster rate, $k_B^- N_B$, proportional to the number $N_B$ of B-cells. As the amount of Ag is depleted, selection of B-cell is more and more stringent, and the GC eventually dies out.

### GC affinity maturation

The GC is initialized with $N_{\text{found}}$ founder clones from the population of naive responders (*Tas et al., 2016*). Their binding energies $\epsilon$ are independently drawn from a Gaussian distribution, with mean $\mu_{\text{naive}}$ and standard deviation $\sigma_{\text{naive}}$ (Histogram one in *Figure 1*). During the initial phase of colonization and expansion, these founder clones duplicate uniformly (without mutation) to produce a population of $N_i$ B-cells. We do not model this initial phase, and start our simulation $T_{\text{GC}}$ days after Ag injection, when the GC is mature (*Victora and Nussenzweig, 2012*; *De Silva and Klein, 2015*).

During each evolution round (of duration $T_{\text{turn}}$ hours), all cells are assumed to divide twice, independently of their affinity. If the number of cells in the population, $N_B$, exceeds some threshold value $N_{\max}$ during the division process, each cell is removed with probability $1 - N_{\max}/N_B$, so that on average only $N_{\max}$ cells are left. Imposing a finite carrying capacity to the GC takes into account limitations on its growth, due to the availability of metabolic resources or the finite amount of T-cell help.

At division B-cells have probability $p_{\text{mut}}$ of developing mutations through a process known as *Somatic Hyper-Mutation*. Mutations can be lethal, neutral, or affinity-affecting with probabilities equal to, respectively, $p_l$, $p_s$, or $p_a$ (*Zhang and Shakhnovich, 2010*). In the latter case, the binding energy of the cell is added a random contribution, $\epsilon \to \epsilon + \Delta\epsilon$, drawn from a log-normal distribution $K_{\text{aa}}(\Delta\epsilon)$ (*Ovchinnikov et al., 2018*), see *Appendix 1—figure 1.A*. Most affinity-affecting mutations are deleterious, that is correspond to $\Delta\epsilon > 0$ (Histogram 2 in *Figure 1*).

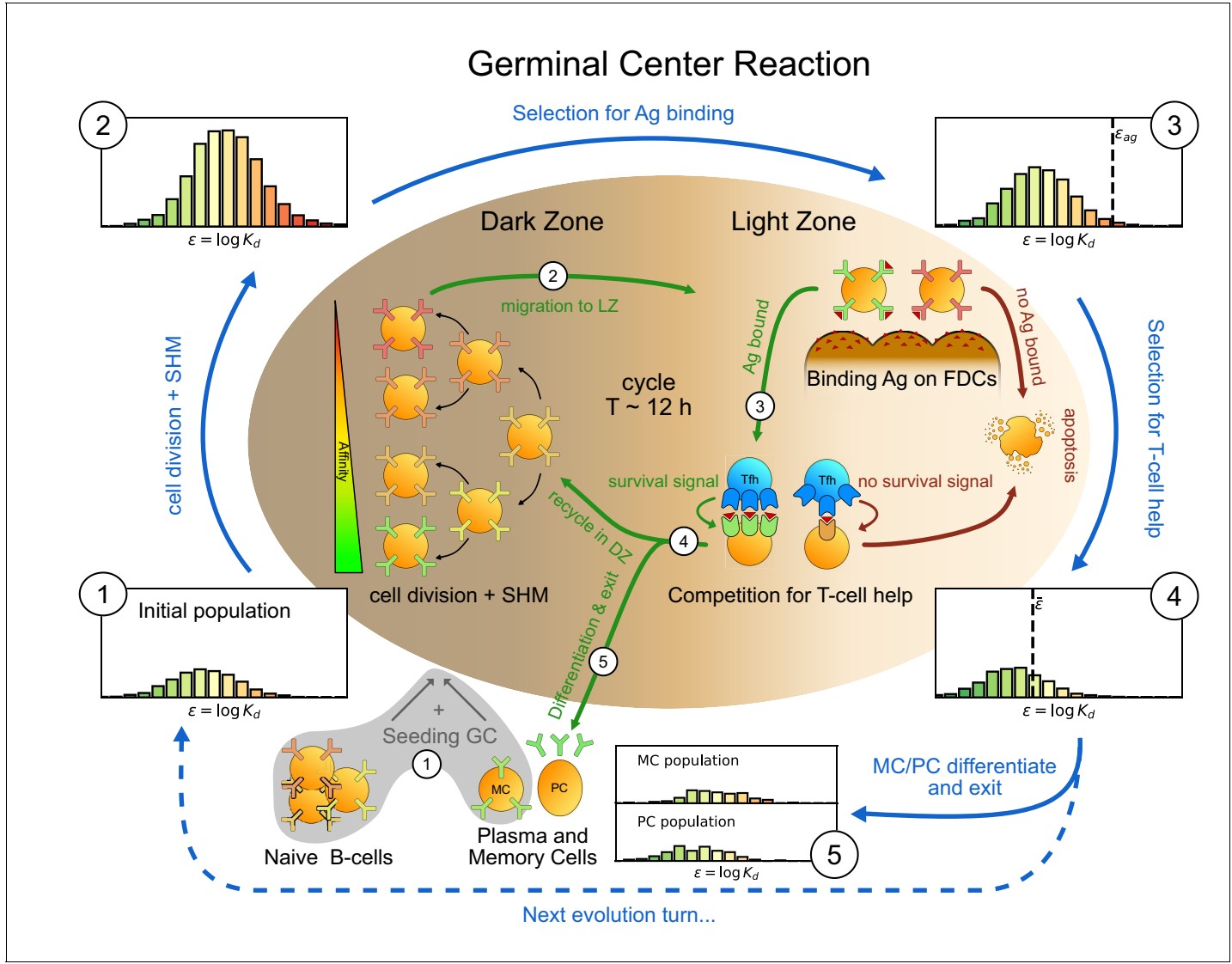

**Figure 1.** Sketch of the germinal center reaction (inner part) and effects of the main reaction steps on the distribution of the binding energies ($\epsilon$, equivalent to the logarithm of the dissociation constant $\log K_d$) of the B-cell population (histograms on the outer part). A red-to-green color-scale is used to depict the affinity of both B-cell receptors in the inner part of the scheme and in the outer binding-energy histograms. Upon Ag administration GCs start to form, seeded by cells from the naive pool having enough affinity to bind the Ag. If the Ag has already been encountered also reactivated memory cells (MC) created during previous GC reactions can take part in the seeding. At the beginning of the evolution round cells duplicate twice in the GC dark zone and, due to somatic hypermutation, have a high probability of developing a mutation affecting their affinity. Most of the mutations have deleterious effects but, rarely, a mutation can improve affinity. As a result the initial population (1) grows in size and decreases its average affinity (2). After duplication cells migrate to the light zone, where they try to bind Ag displayed on the surface of follicular dendritic cells. Failure to bind Ag eventually triggers apoptosis. The probability for a cell to successfully bind the Ag depends both on its affinity for the Ag and on the amount of Ag available. Cells with binding energy higher than a threshold value $\epsilon_{Ag}$ are stochastically removed (3). The Ag concentration shifts this threshold by a quantity $\log C$. B-cells able to bind the Ag will then internalize it and display it on MHC-II complexes for T-cells to recognize, and then compete to receive T-cell help. We model this competition by stochastic removal of cells with binding energy above a threshold $\bar{\epsilon}$ that depends on the affinity of the rest of the population (4). As before Ag concentration shifts this threshold. Moreover to account for the finite total amount of T-cell help available we also enforce a finite carrying capacity at this step. Surviving cells may then differentiate into either MC that could seed future GCs or Ab-producing plasma cells (PC). MCs and PCs are collected in the MC/PC populations (5), while the rest of non-differentiated cells will re-enter the dark zone and undergo further cycles of evolution. Eventually Ag depletion will drive the population to extinction.

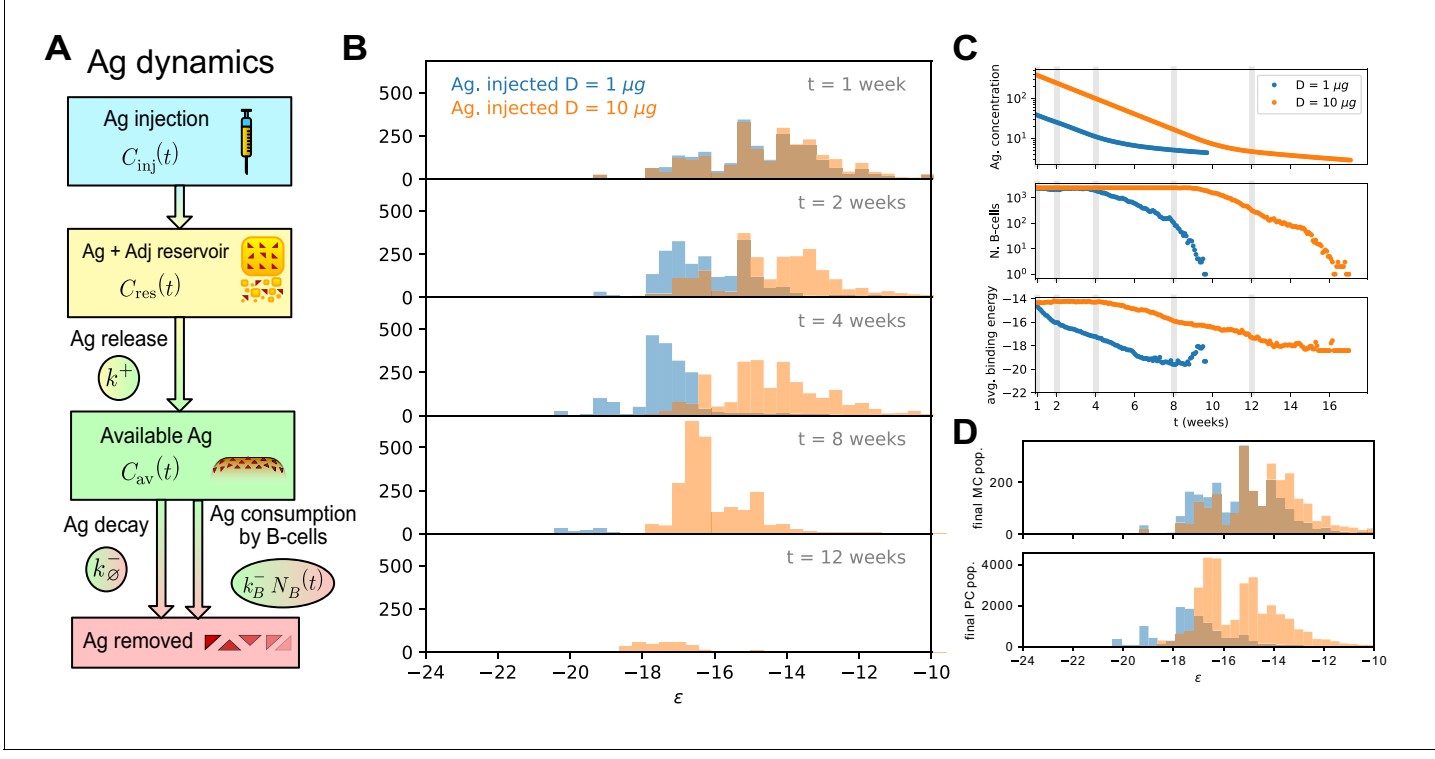

**Figure 2.** Effect of different antigen dosages on model evolution. (**A**) Schematic representation of the Antigen (Ag) dynamics. Upon injection Ag is added to the reservoir. From there it is gradually released at a rate $k^+$ and becomes available for B-cells to bind. Available Ag is removed through decay at a constant slow rate $k_\emptyset^-$ and consumption by the B-cells at rate $k_B^- N_B$, proportional to the size of the B-cell population. (**B**) Histogram of the B-cell populations at different times (1,2,4,8,12 weeks after Ag administration) for two simulations of the model at two different values of administered Ag dosage (1 $\mu$g - blue, 10 $\mu$g - orange). Ag Dosage $D$ is converted to Ag concentration $C$ through the inferred proportionality constant $\alpha = D/C = 23$ ng. Notice that low dosage entails a faster maturation, albeit having a shorter total duration. (**C**) Evolution of Ag concentration (top), number of B-cells in germinal center (middle) and average binding energy of the population (bottom) for the same two simulations as a function of time from Ag administration. Vertical grey lines corresponds to time points for which the full affinity distribution is displayed in panel (**B**). (**D**) cumulative final populations of differentiated cells at the end of evolution (memory cells - top, plasma cells - bottom) for the two simulations. Colors encode Ag dosage as in panel B and C. Simulations were performed with variant (**C**) and parameters given in *Table 1*.

After duplication B-cells are first selected according to their capability to bind Ags exposed on FDCs (*Figure 1* top right). The probability for a cell to survive this selection step is a decreasing function of its binding energy $\epsilon$ and increases with the concentration $C$ of Ag on FDCs; it is given by

$$P_{\mathrm{Ag}}(\epsilon) = \frac{Ce^{-\epsilon}}{Ce^{-\epsilon} + e^{-\epsilon_{\mathrm{Ag}}}} \, , \qquad (1)$$

where $\epsilon_{\mathrm{Ag}}$ is a threshold binding energy (*Appendix 1—figure 1C*). As a consequence, cells with high binding energy (larger than $\epsilon_{\mathrm{Ag}} + \log C$) are likely to be removed from the population, compare Histograms 2 and 3 in *Figure 1*.

Following internalization, B-cells load the Ag on MHC molecules on their surface (*Nowosad et al., 2016*; *Natkanski et al., 2013*; *Batista and Neuberger, 2000*). By probing these molecules T follicular helper cells provide survival signals to the B-cells with high Ag affinity (*Figure 1*, 'Competition for T-cell help') (*Allen et al., 2007b*; *Shulman et al., 2014*; *Victora and Nussenzweig, 2012*; *Depoil et al., 2005*). The probability that a B-cell with binding energy $\epsilon$ survives this second step of selection is

$$P_{\mathrm{T}}(\epsilon, \bar{\epsilon}) = a + (1 - a - b)\frac{Ce^{-\epsilon}}{Ce^{-\epsilon} + e^{-\bar{\epsilon}}} \, , \quad \text{with} \quad e^{-\bar{\epsilon}} = \langle e^{-\epsilon} \rangle_{GC} . \qquad (2)$$

The threshold energy $\bar{\epsilon}$ depends on the current state of the B-cell population in the GC, as a result of the competition amongst these cells for getting the survival signal from T-helper cells, see

**Table 1.** List of parameters in the model and of their values.

Binding energies are expressed in units of $k_B T$, and times in days (d) or hours (h). The last nine parameters were inferred within selection variant (C), except $\epsilon_{\text{Ag}}$, whose reported value refers to variant (A), which includes Ag-binding selection.

**Values of model parameters**

| Symbol | Value | Meaning | Source |
|---|---|---|---|
| $T_{\text{turn}}$ | 12 h | Duration of an evolution turn | *Wang et al., 2015* |
| $T_{\text{GC}}$ | 6 d | Time for GC formation after injection | *De Silva and Klein, 2015*; *Jacob et al., 1993*; *McHeyzer-Williams et al., 1993* |
| $N_{\text{max}}$ | 2500 | GC max population size | *Eisen, 2014*; *Tas et al., 2016* |
| $N_i$ | 2500 | Initial GC population size | *Eisen, 2014*; *Tas et al., 2016* |
| $N_{\text{found}}$ | 100 | Number of GC founder clones | *Tas et al., 2016*; *Mesin et al., 2016* |
| $p_{\text{diff}}$ | 10% | Probability of differentiation | *Wang et al., 2015*; *Meyer-Hermann et al., 2012*; *Oprea and Perelson, 1997* |
| $\tau_{\text{diff}}$ | 11 d | Switch time in MC/PC differentiation | *Weisel et al., 2016* |
| $\Delta\tau_{\text{diff}}$ | 2 d | Switching timescale in MC/PC differentiation | *Weisel et al., 2016* |
| $p_{\text{mut}}$ | 14% | Prob. of mutation per division | *Wang et al., 2015*; *McKean et al., 1984*; *Kleinstein et al., 2003* |
| $p_{\text{s}}, p_{\text{l}}, p_{\text{aa}}$ | 50%, 30%, 20% | Probability of a mutation to be silent/lethal/affinity-affecting | *Zhang and Shakhnovich, 2010*; *Wang et al., 2015*; *Wang, 2017* |
| $K_{\text{aa}}(\Delta\epsilon)$ | *Equation 18* | Distribution of affinity-affecting mutations | *Ovchinnikov et al., 2018* |
| $k^+$ | 0.98 /d | Ag release rate | *MacLean et al., 2001* |
| $k_{\emptyset}^-$ | $1.22 \times 10^{-2}$/d | Ag decay rate | *Tew and Mandel, 1979* |
| $a$ | 0.12 | Baseline selection success probability | Max-likelihood fit |
| $b$ | 0.66 | Baseline selection failure probability | Max-likelihood fit |
| $\mu_{\text{naive}}$ | -14.60 | Mean binding energy of seeder clones generated by naive precursors | Max-likelihood fit |
| $\sigma_{\text{naive}}$ | 1.66 | Standard deviation of the seeder clones binding energy distribution | Max-likelihood fit |
| $k_B^-$ | $2.07 \times 10^{-5}$ /d | Ag consumption rate per B-cell | Max-likelihood fit |
| $\alpha$ | $2.3 \times 10^{-2}\,\mu g$ | Concentration to dosage conversion factor | Max-likelihood fit |
| $g_{\text{recall}}$ | 0.56 | MC fraction in Ab-SC population for measurement 1 day after boost | Max-likelihood fit |
| $g_{\text{imm}}$ | 0 | MC fraction in Ab-SC population for measurement 4 days after second injection | Max-likelihood fit |
| $\epsilon_{\text{Ag}}$ | -13.59 | Threshold Ag binding energy [(A)] | Max-likelihood fit |

Histogram four in *Figure 1*. Parameter $a$ represents the probability for any B-cells to be selected due to stochastic effects (e.g. bystander activation [*Horns et al., 2019*]) even with very low affinity; it is introduced to reproduce the observation that selection in GCs is permissive in the presence of complex Ags such as the ones found in vaccines (*Finney et al., 2018*). Parameter $b$ instead represents the probability for a B-cell to fail selection at high affinity. The introduction of $b$ comes from the experimental observation that part of the population of apoptotic cells in GCs has high affinity for the antigen (*Mayer et al., 2017*); the removal of these cells could result from stochastic effects (*Lau and Brink, 2020*).

We will consider three variants of the above selection process: (A) two-step selection described in *Equations 1 and 2*; (B) same two-step selection, but without permissiveness, that is with $a = b = 0$; (C) simpler selection process based on competition for T-cell help only, that is *Equation 2*, but allowing for permissiveness.

## Differentiation into plasma and memory cells

Clones that successfully survive selection differentiate with probability $p_{\mathrm{diff}}$ in either Ab-producing plasma cells (PCs) or long-lived memory cells (MCs), or start a new evolution cycle with probability $1 - p_{\mathrm{diff}}$. The probabilities of differentiation into MC and PC, respectively, $\mu_{\mathrm{MC}}(t)$ and $\mu_{PC}(t) = 1 - \mu_{\mathrm{MC}}(t)$, depend on the time following Ag injection $t$ (early vs. late response) (**Weisel et al., 2016**). The MC cell fate is more likely at the beginning of evolution and the PC is more likely towards the end, effectively resulting in a temporal switch occurring around day 11 after injection (**Weisel et al., 2016**; **Appendix 1—figure 1B**). The MC and PC populations (Histograms five in **Figure 1**) grow at each evolution step, as more and more clones differentiate.

Administering a recall Ag injection some time after vaccination generates responders Ab-secreting cells (Ab-SCs). These cells comprise both MCs, that can be stimulated to differentiate and produce Abs upon new Ag encounter (**McHeyzer-Williams et al., 2015**; **Dogan et al., 2009**; **Mesin et al., 2016**; **Inoue et al., 2018**), and residual PCs formed during previous maturations; PCs belonging to the long-lived pool are capable of surviving up to a human lifetime in the absence of division (**Wong and Bhattacharya, 2019**; **Crotty et al., 2003**). The affinity distribution of Ab-SCs is assumed to be a weighted mixture of the MC and PC populations, with fractions equal respectively to $g$ and $1 - g$, where the value of $g$ is expected to depend on the conditions under which the system is probed.

Cells harvested from the spleen originate from multiple GCs. To account for this phenomenon, we carry out several parallel stochastic simulations of GCs ($N_{\mathrm{GCs}} = 20$); the GCs are initialized with different populations of founders, and produce different Ab-SC populations. The distribution of affinities, averaged over the GCs, is our outcome and can be compared to experimental results. We choose not to introduce interactions between the evolving GCs, due to the lack of experimental quantification of possible GC-crosstalk.

## GC reinitialization

When a second Ag injection is performed after the end of the first GC reaction a new GC is initiated. The population of $N_{\mathrm{found}}$ founder clones for the new GC is composed of both new GC B-cells with naive precursors having sufficient affinity to bind the Ag, and reactivated MCs accumulated during the past evolution (**McHeyzer-Williams et al., 2015**; **Dogan et al., 2009**; **Inoue et al., 2018**). The probability for a founder cell to be extracted from the MC pool is $p_{\mathrm{mem}} = N_{\mathrm{mem}}/(N_{\mathrm{mem}} + N_i)$, where $N_{\mathrm{mem}}$ is the number of MCs accumulated up to the time of the second injection. This hypothesis reflects the fact that we expect more reactivated MCs to colonize the newly formed GC if more MCs were produced in the previous maturation. However, one could also consider this ratio to be constant (see appendix sect. 6 'Possible model variations').

If the Ag injection occurs before the end of the first GC reaction, only the MC produced so far are considered to seed the second GC reaction. This initial exchange of MCs is the only interaction between the two GCs, which evolve independently at later times.

## Values of model parameters

The values of all but nine model parameters listed above were extracted from existing literature, see description in appendix section (Model definition and parameters choice) and table of parameter values in **Table 1**. The remaining nine parameters, which were either not precisely known or strongly dependent on our experimental protocol, were fitted from the experimental data through a Maximum-Likelihood inference procedure for each selection variant (A), (B) or (C); the inference procedure is described in Materials and methods and in appendix sect. 4. These fitted parameters describe: the initial distribution of affinities ($\mu_{\mathrm{naive}}$, $\sigma_{\mathrm{naive}}$), the Ag-binding selection threshold ($\epsilon_{\mathrm{Ag}}$, not included in variant (C)), the Ag-comsumption rate per B-cell ($k_B^-$), the permissiveness characterizing parameters ($a$, $b$, not included in variant (B)), the contribution $g$ of MC to Ab-SC population (for the 1- and 4-day protocols in our experiments), and the conversion factor $\alpha$ between vaccine Ag dosage $D$ in units of mass and dimensionless injected concentration $C^{\mathrm{inj}}$ (we express $\alpha$ as a mass, which makes concentrations dimensionless).

## Phenomenology of the stochastic affinity maturation model

### Schematic evolution of the affinity distribution in the course of maturation

In *Figure 2 B, C and D* we report the result of two stochastic simulations of our model on a protocol consisting of a single Ag injection. The simulations differ by the administered Ag dosages $D = 1$ (blue) or 10 (orange) $\mu$g. The founder clones population is the same in the two simulations in order to eliminate differences coming from variations in the affinities of the initial population. For both concentrations, the main phases in the evolution of the GC can be summarized as follows. After injection and before the start of the GC reaction at day six the amount of available Ag increases due to gradual release from the adjuvant matrix, while consumption exponentially increases. At the beginning of the simulation (day 6) the GC is at maximum capacity and the driving contribution to Ag depletion is consumption by B-cells, which occurs at a rate $k_B^- N_{\max}$ (*Figure 2C*). This consumption continues until Ag concentration reaches a critical value, at which selection pressure becomes strong enough to reduce the population size (despite the duplication step) and eventually drives GCs to extinction (*Figure 2C*).

### Maturation induces progressive loss of clonality

We investigated how the changes in affinity reflect changes in the clonal population in the GC. Recent experiments *Tas et al., 2016*; *Abbott et al., 2018*; *Kuraoka et al., 2016* have shown that maturation is accompanied by various degrees of homogenizing selection, that is, a reduction of clonality, leading in some cases to strong clonal dominance. We assess the impact of homogenizing selection in our model by keeping trace of the offspring of each founder clone in the stochastic evolution of a single GC for a 4 weeks time-span. The evolution of clonality is reported for two representative simulations in *Figure 3A, B*. The plot report the cumulative composition of the population as a function of time; the offspring of each founder clone is represented by a different color, associated to the binding energy of the founder clone, see color scale on the right. In the simulation reported in *Figure 3A*, a single clonal family ensued from a high-affinity clone progressively expands, and constitutes around 70% of the total GC population at 4 weeks. In the simulation reported in *Figure 3B*, no clone dominates the population, and the GC maintains its polyclonality throughout maturation, with many good affinity clones sharing substantial fractions of the GC.

To quantify the evolution of homogenization over time, we estimated the fraction of the population constituted by the most expanded clone at each given time, where 100% would correspond to the GC being completely populated by the offspring of a single founder clone. In *Figure 3C*, we plot the distribution of this most-expanded-clone fraction 1000 stochastic simulations at four different time-points (1,2,3,4 weeks after injection). All GCs in our simulations are highly polyclonal at the beginning, with each clone constituting 1% of the initial population. As time goes on, however, more and more GCs feature a dominant clone, sometimes with a very high population fraction. The median of the frequency distribution at week 4 is around 30%, meaning that in half of the simulated GCs a single clonal family makes up for more than 30% of the total B-cell population. Finally, in *Figure 3D* we plot, for each simulation, the final (week 4) fraction of the population corresponding to the most-abundant clonal family against its initial binding energy. As expected homogenization correlates with the presence of a high-affinity founder precursor.

### Efficacy of affinity maturation varies non monotonically with Ag dosage

Inspection of *Equations 1 and 2* shows that the role of Ag concentration in our model is to shift the selection thresholds by $\log C$. This shift has two different consequences. First, its affects the speed of affinity maturation, that is, the decrease in the population average binding energy per round of evolution. The histograms in *Figure 2B* (area reflects the size of the population) and the curve for the average binding energy of the population in *Figure 2C* show that smaller Ag dosages correspond to faster affinity maturation. Secondly, strong or weak selection resulting from, respectively, small or large concentrations also affects the changes over time in the size of the B-cell population, which in turns impacts the Ag-consumption rates and, therefore, the lifetime of the GC. This can be again visualized by comparing population evolutions on the histograms of *Figure 2B* and on the curve of the population size in *Figure 2C*.

These two competing effects concur to shape the final MCs and PCs binding energy distribution (*Figure 2D*). Protection against future pathogen encounters will be granted by these cells, and as

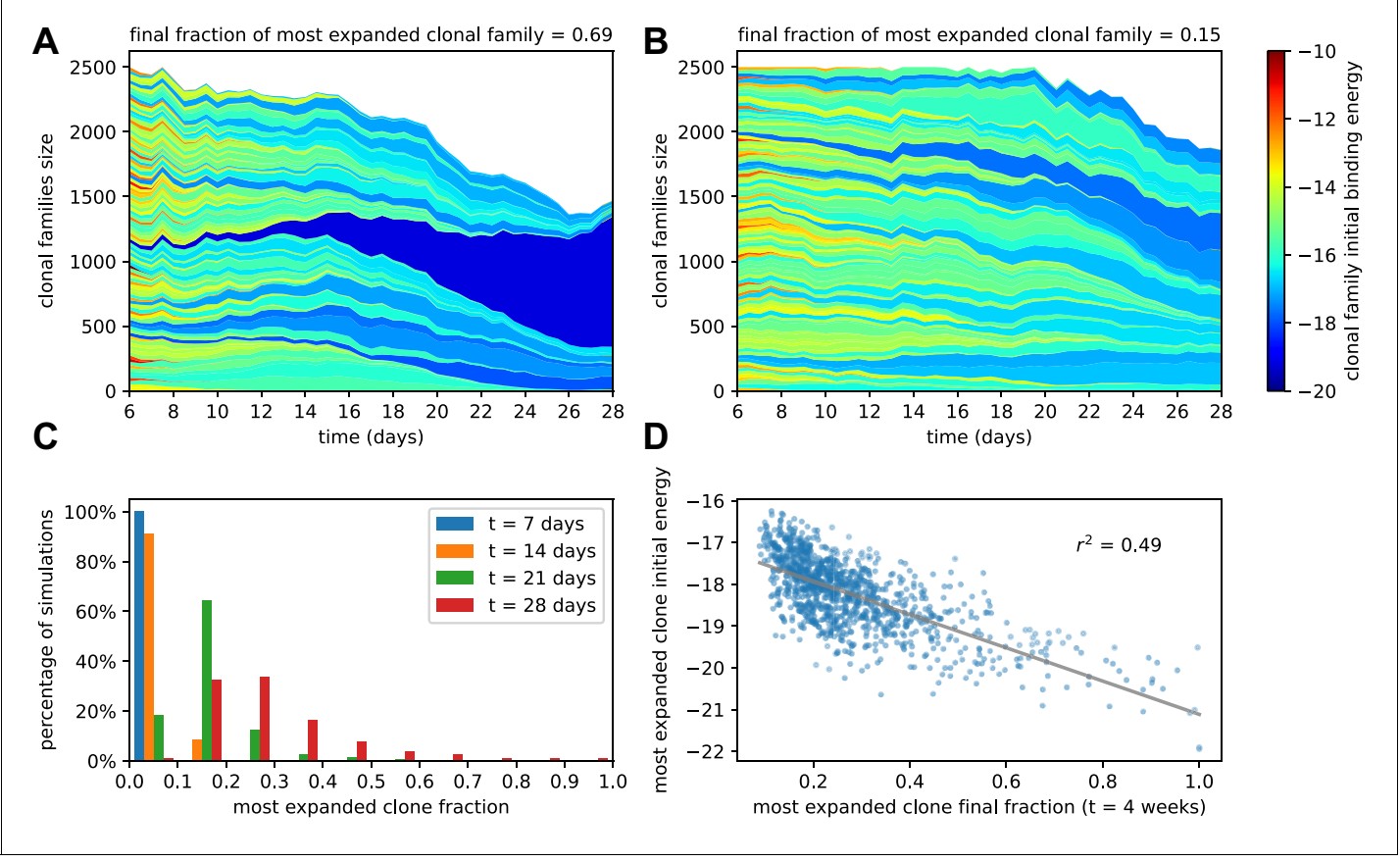

**Figure 3.** Simulated GCs present different levels of homogenization. (**A**) Example of homogenizing selection in GC evolution. Population size as a function of time for each clonal family in stochastic simulations of a single GC. The GC were initiated with an injected antigen dosage of $D = 1\mu g$. The color of the clonal family reflects the initial binding energy of the founder clone according to the color-scale on the right. On top, we report the fraction of the final population composed by the most expanded clonal family. In this example, the progeny of a single high-affinity founder clone (dark blue) progressively takes over the GC, and at week 4 constitutes around 70% of the GC B-cell population. (**B**) Example of heterogeneous GC evolution. Contrary to the previous example, many clonal families coexist, without one dominant clone taking over the GC. (**C**) Evolution of the distribution of the most-expanded clone fraction. We perform 1000 stochastic simulations and evaluate the fraction of the population constituted by the most-expanded clone at each time (cf colors in the legend). Distributions show the percentage of simulations falling in 10 bins splitting equally the [0,1] interval according to the values of their dominant clone fractions. Notice the presence of heterogeneous and homogeneous GCs at week 4. (**D**) Scatter plot of final (week 4) population fraction versus initial binding energy for the most-expanded clone; the straight line shows the best linear fit ($r^2 \simeq 0.49$). The presence of a clone with high initial affinity favors the advent of a homogeneous GC.

such their affinity distribution can be used as an indicator to estimate the success and quality of the immunization procedure.

Because of the double role that Ag concentration plays in controlling the maturation rate and the duration of the GC reaction, in our model the optimal average binding energy of the MC and PC population is achieved at intermediate Ag dosages. Intuitively, this can be explained by observing that, while small Ag dosages cause faster affinity gains, they also result in fast population decrease and short maturation. Therefore in this scenario only a few high-affinity cells will be produced. Conversely, if the dosage is too high then a lot of mediocre or intermediate affinity clones will accumulate, and the high-affinity clones obtained at the end of the evolution process will be in minority. Only intermediate dosages realize a good combination of good maturation speed and population survival. In order to better understand this phenomenon, we can introduce a deterministic version of the model, which is both able to reproduce the average of stochastic simulations and is also amenable to detailed mathematical analysis.

## Resolution of the model offers insight on effect of ag dosage

### Deterministic evolution reproduces stochastic simulations

In order to gain insight on the non-monotonic effects of concentration onto affinity maturation, we introduce a deterministic version of the model, which formally becomes exact in the limit of very large sizes $N$. In practice, when the size of the population is big enough, the distribution of binding energies can be considered as continuous. The evolution of this continuous distribution $\rho(\epsilon, t)$ over time (number of rounds) $t$ becomes deterministic (Materials and methods); in other words, the stochastic nature of the underlying process disappears in this limit. This introduces a twofold advantage. Firstly, studying deterministic rather than stochastic evolution is a significant simplification, which allows mathematical analysis, see section (Theoretical analysis at fixed concentration) Secondly, numerically evaluating the average outcome of an immunization scheme is computationally much cheaper if done through the deterministic model rather than by averaging many stochastic simulations. This is of paramount importance when using our stochastic fitting procedure, which requires simulating the system for many different values of the parameters.

As a first check, we compare the predictions of the deterministic solution of the model with the corresponding averages for the stochastic simulations to verify that they are in good agreement. For example in *Apendix 1—figure 2A-D*, we show the size of the GC B-cell population, and the average binding energies for the GC B-cell, MC and PC populations, averaged over 1000 simulations, which are in very good agreement with their theoretical counterparts. Notice that the model looses accuracy when the population size is too small (cf accuracy of predictions for GC B-cells average binding energies in *Apendix 1—figure 2C*), as expected. However, these finite-size effects are generally irrelevant, since low-population size states contribute only marginally to the final MCs/PCs distributions we are interested in (cf accuracy of predictions for MCs and PCs average binding energies in *Apendix 1—figure 2B and D*). The deterministic theory is therefore able to accurately predict the full Ab-SC distributions (see *Apendix 1—figure 7*, blue distributions correspond to the deterministic solution, and green histograms to the average distribution over 1000 stochastic simulations).

### Theoretical analysis at fixed concentration

We can gain deep insight on the role of Ag concentration in regulating maturation by studying the theoretical solution of the model in the special case of constant Ag concentration $C$. To be able to observe asymptotic population expansion, we momentarily relax the maximum population size constraint, and set $N_{\max} = \infty$. Furthermore, for variants (A) and (B), we consider that the cells in the population have high enough affinity to successfully overcome the first selection step, that is $\epsilon \ll \epsilon_{\mathrm{Ag}}$; this assumption is not necessary for variant (C), which does not include Ag-binding selection. The effects of these simplifications will be discussed below.

In *Figure 4A*, we report the evolution of the distribution of binding energies with constant Ag concentration $C = 30$ (top right). Notice that the distribution is not normalized to one, but to the number of cells in the population. Color encodes time from the beginning of the GC evolution. We observe that, as the number $t$ of evolution rounds increases, the size of the population increases exponentially with a growth rate $\phi$ (top left) and the average binding energy shifts linearly, with a speed $u$ (bottom left). The distribution of binding energies therefore evolves as a travelling wave of profile $\rho^*$, with exponentially increasing size:

$$\rho(\epsilon, t) \simeq \exp\{\phi\, t\} \times \rho^*(\epsilon - u\, t)\,. \tag{3}$$

This behavior can be mathematically established, and the growth rate $\phi$ and maturation speed $u$ computed by solving an appropriate eigenvalue equation. To do so, we introduce the evolution operator $\mathbf{E}$ that describes how the distribution of binding energies evolves after each round of maturation. Briefly speaking, $\mathbf{E}(\epsilon, \epsilon')$ is the average number of B-cells with energy $\epsilon$ produced, through the duplication, selection and mutation steps by an ancestor cell of energy $\epsilon'$ (Materials and methods); it depends on the Ag concentration $C$ through the selection step, see *Equation 2*. The travelling wave behavior for the distribution of binding energies expressed in *Equation 3* implies that

$$e^\phi\, \rho^*(\epsilon) = \int d\epsilon'\, \mathbf{E}(\epsilon + u, \epsilon')\, \rho^*(\epsilon')\,. \tag{4}$$

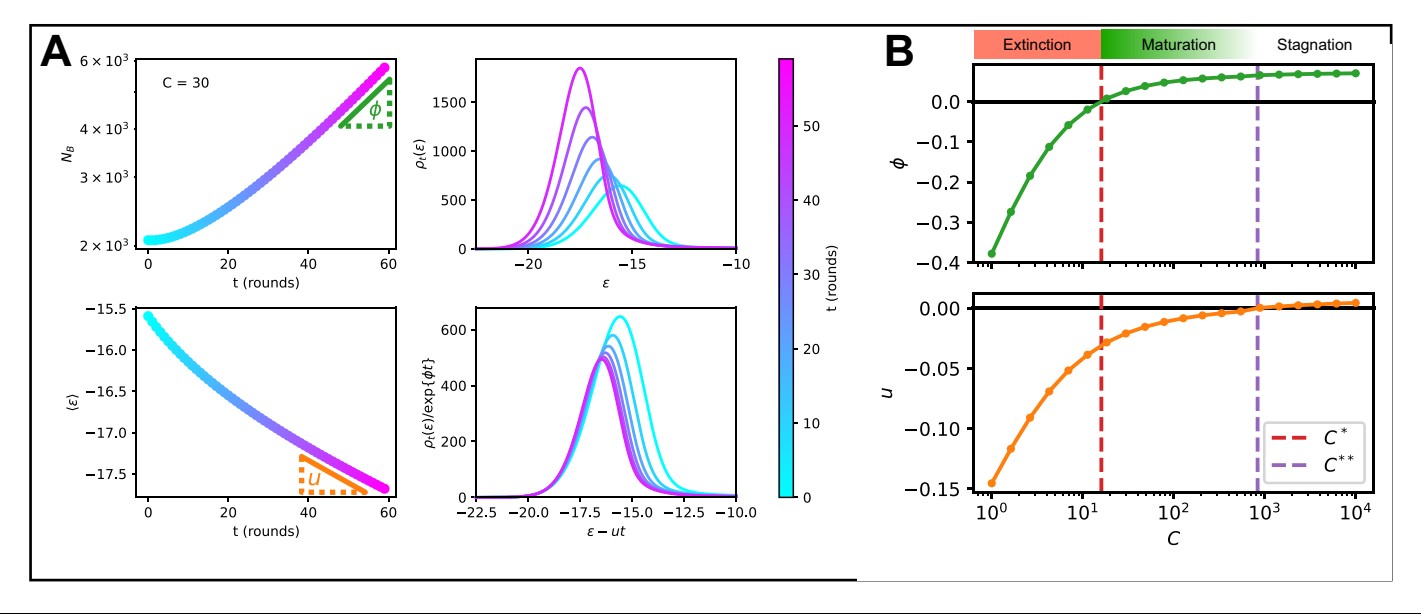

**Figure 4.** Asymptotic evolution at constant Ag concentration. (**A**) Analysis of the asymptotic deterministic evolution for the large-size limit of the model, at constant available concentration $C = 30$. Top left: size of the population vs. number of maturation rounds, showing the exponential increase at rate $\phi$. Bottom left: average binding energy of the B-cell population, decreasing linearly with speed $u$. Top right: evolution of the binding energy distribution, normalized to the number of cells in the GC, shows a travelling-wave behavior. Different times are represented with different colors, according to the color-scale on the right. Bottom right: distributions of binding energies, shifted by the time-dependent factor $-ut$ and rescaled by the exponential factor $\exp\{-\phi t\}$. Notice the convergence to the invariant distribution $\rho^*$. (**B**) Values of the growth rate $\phi$ (top) and maturation speed $u$ (bottom) as functions of the Ag concentration $C$. The points at which the two quantities are zeros define the two critical concentration $C^*$ and $C^{**}$ (red and purple vertical dashed lines). They split the asymptotic behavior of the system at constant Ag concentration in three different regimes: extinction for $C<C^*$ ($\phi<0$), maturation for $C^*<C<C^{**}$ ($\phi>0$ and $u<0$) and finally stagnation for $C>C^{**}$ ($\phi>0$ but $u\geq0$). Results were obtained using parameter values reported in *Table 1*.

This eigenvalue equation can be solved to determine the growth rate $\phi$, the wave (maturation) speed $u$, and the wave profile $\rho^*$ as functions of the concentration $C$. More details on eigenvalue equation *Equation 4* and on how it can be numerically solved can be found in Materials and methods Eigenvalue equation and phase diagram and Appendix (Theoretical solution and eigenvalue equation).

Results are shown in *Figure 4B*. Two special values of the concentration are $C^*$, the concentration at which the growth rate $\phi$ vanishes, and $C^{**}$, the concentration at which the maturation speed $u$ vanishes. Distinct regimes of maturation are found, depending on the dosage $C$:

- At low Ag concentration $C<C^*$, both $\phi$ and $u$ are negative: the strong selection pressure produces high affinity clones and maturation is fast, but the number of cells decreases exponentially, leading to a quick extinction of the population.
- At high concentration $C>C^{**}$, the selection pressure is too weak to compensate the deleterious drift due to mutations, and binding energies increase on average at each round ($u>0$). The growth rate $\phi$ is positive, hence an exponentially increasing number of poor-quality B-cells are produced.
- In the intermediate range of concentration, $C^*<C<C^{**}$, we have both population expansion (positive growth rate $\phi$) and affinity maturation (negative maturation speed $u$). The most efficient maturations are obtained for values of $C$ slightly exceeding $C^*$, as $u$ is very close to 0 for values of $C$ tending to $C^{**}$ (*Figure 4B*).

The above analysis provides a detailed picture of the effect of Ag concentration on population growth and maturation, even when realistic constraints are reintroduced. First, if we forbid the population to expand indefinitely and enforce the maximum carrying capacity ($N_{max}$) again, the value of $u(C)$ is not modified, since this constraint has no effect on affinity. It also does not influences the regime $C<C^*$ in which the population contracts ($\phi(C)<0$). However, it prevents the population from

expanding, thus setting effectively the maximum asymptotic growth rate to $\phi(C) = 0$ if $C \geq C^*$. Second, if we reintroduce Ag-binding selection we observe no difference in asymptotic behavior when the population is maturing ($C < C^{**}$ and $u < 0$). However for high concentration $C > C^{**}$ a positive asymptotic velocity is not possible, since in this case the distribution will eventually reach the threshold Ag-binding energy and this selection will prevent further affinity decrease. This limits the maximum asymptotic velocity to 0 and maximum growth rate to $\phi(C^{**})$. Finally, when the Ag concentration is not kept constant but varies during immunization through consumption and decay (*Figure 2A*), the maturation behaviors observed during GC evolution (*Figure 2B and C*) can be understood depending on whether the value $C$ of the concentration crosses the boundaries $C^{**}$ or $C^*$ over time.

## Model distributions of affinities match experimental measurements in immunized mice

### Probing immunization outcome through single-cell affinity measurements

We compare our model predictions for the effects of Ag dosage and release schedule to experimental data from mice immunization against TT (Materials and methods). These data consist of single-cell affinity measurements performed on IgG Secreting Cells (IgG-SCs) extracted from mice spleen following immunization. In practice, we immunize mice according to different immunization schemes, described below. Following immunization cells from the spleen are harvested, purified and the affinity of single IgG-SCs is measured according to the protocol developed in *Eyer et al., 2017*. By pooling all the measurements from mice immunized according to the same scheme we are able to obtain a full affinity distribution, such as the ones reported in *Figure 5* (orange histograms, for each histogram the number of mice and pooled measurements is indicated). Measurements are limited by experimental sensitivity. In particular, only affinities above the minimum affinity limit of $K_d = 500\,\mathrm{nM}$, that is energies below $\epsilon_{\max} = -23.03$ are measurable. In addition, our measurement technique cannot resolve affinities higher than $K_d = 0.1\,\mathrm{nM}$. The range of energies accessible to measurements is represented with the gray shaded area in the histograms of *Figure 5*. These distributions give us an affinity snapshot of the Ab-producing cell population; they contain much more information than average quantities, such as the average serum affinity. Our approach allows to probe both tails of the affinity spectrum, and to fully test the effectiveness of the immunization procedure. In our experiments we test three different immunization protocols, schematized in *Figure 5* (top row). Scheme 1 consists of two injections of a dose $D$ of Ag, separated by a 4 weeks interval. Cells are harvested 4 days after the second injections. In the first injection, the Ag is mixed with Complete Freund's Adjuvant (CFA), whilst in the second Incomplete Freund's Adjuvant (IFA) is used. In this protocol, we tested five different Ag dosages: $D = 0.01, 0.1, 0.5, 1$ and $10\,\mu\mathrm{g}$ TT. Only four of them are reported in *Figure 5* but the rest can be found in *Appendix 1—figure 7*. Scheme 2 (see *Figure 5* middle column) is identical to scheme one up to the second injection. At this point, after an additional 4 weeks delay, a boost injection of $1\,\mu\mathrm{g}$ pure TT is administered and cells are harvested 1 day later. Tested dosages are $D = 0, 0.01, 0.1, 0.5, 1, 3$ and $10\,\mu\mathrm{g}$ TT. Finally, scheme 3 (see *Figure 5* right column) is the same as scheme two with a differences. Instead of varying the injected Ag dosage, which is kept constant at $D = 10\,\mu\mathrm{g}$ TT, in this scheme the time delay between the first two injections $\Delta T$ is varied. We test four different values for this delay: $\Delta T = 1, 2, 4$ and 8 weeks. These protocols have also been used in *Eyer et al., 2020*, and thanks to the multiple injections they allow us to study the effect of memory recall in subsequent immunizations.

### Inference of model parameters and match with full experimental affinity distributions

We now use the full variety of the data (distributions of affinities obtained with different immunization protocols) to compute and maximize the likelihood of the model as a function of the parameter values. Our objective is two-fold. First, we expect the inferred parameters to provide insights on hardly measurable features of AM, in particular, on the complex steps of selection in our model. Secondly, we show that a single set of parameters is able to accurately reproduce all the experimental measurements corresponding to different situations.

We have implemented a version of the *Parallel Tempering* algorithm (*Swendsen and Wang, 1986*; *Sugita and Okamoto, 1999*) to perform a stochastic search in parameter space and

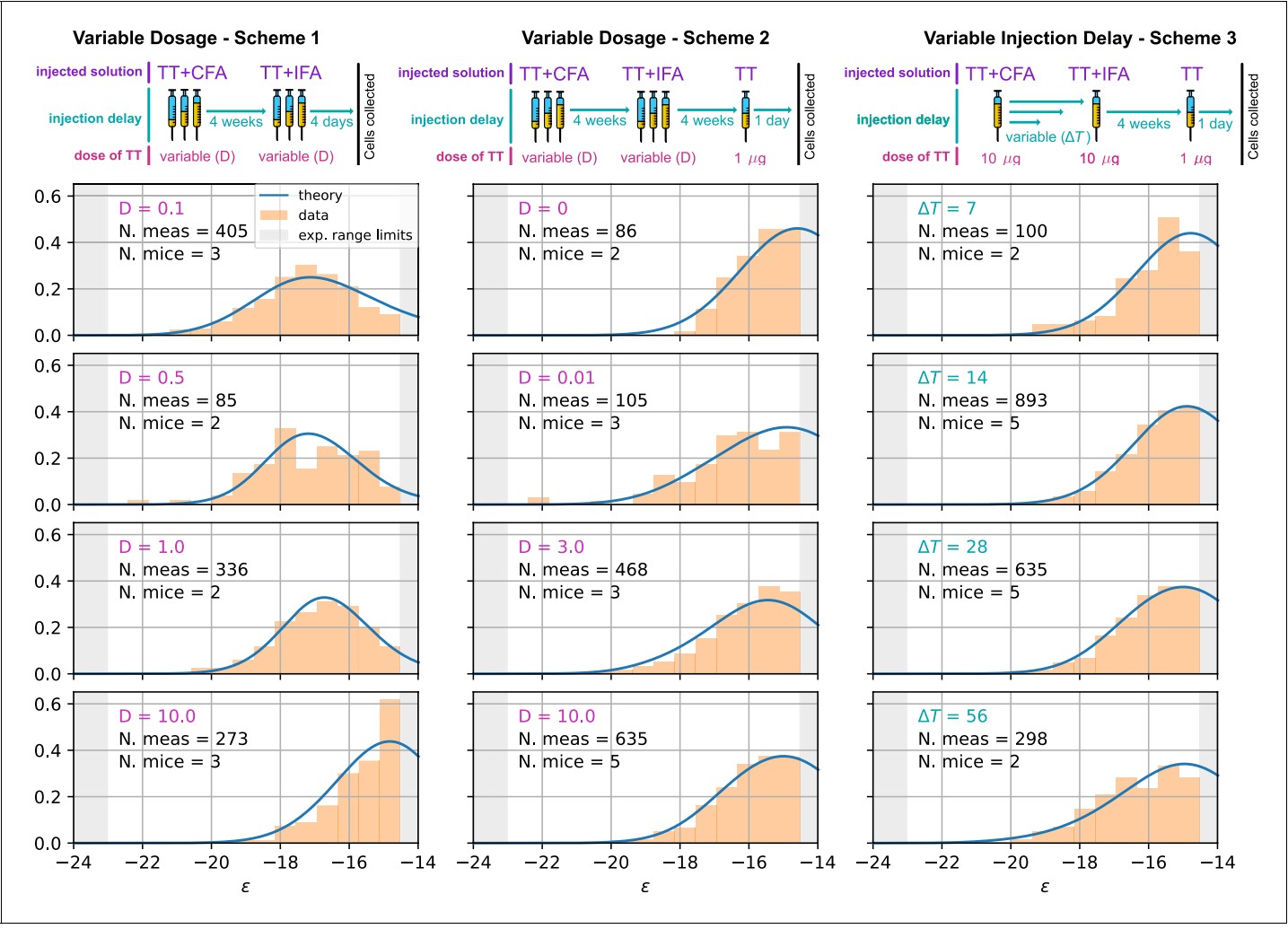

**Figure 5.** Comparison between model-predicted and experimentally measured affinity distributions of antibody-secreting cells (Ab-SCs) for different immunization protocols. A schematic representation of the protocol used is reported on top of each column. Scheme 1 (left column) consists of two injections at the same Ag dosage $D$, separated by a 4 weeks delay. Cells are harvested 4 days after the second injection. Scheme 2 (middle column) is the same as scheme one until the second injection. Then, after an additional 4 weeks delay, a supplementary boost injection of 1μg pure TT is administered, and cells are harvested one day later. Scheme 3 (right column) is the same as Scheme two but the TT-dosage $D = 10\mu g$ of the first two injections is kept constant, and instead the delay $\delta T$ between them is varied. Experimental data (orange histograms) consists in measurements of affinities of IgG-secreting cells extracted from mice spleen. The experimental sensitivity range ($0.1\,\mathrm{nM} \leq K_d \leq 500\,\mathrm{nM}$, or equivalently $-23.03 \geq \epsilon \geq -14.51$) is delimited by the gray shaded area. Blue curves represent the expected binding energy distribution of the Ab-SCs population according to our theory under the same model conditions. For a good comparison, all the distributions are normalized so that the area under the curve is unitary for the part below the experimental sensitivity threshold. For every histogram, we indicate the number of single cell experimental measurements that make up the experimental distribution (black), the number of different mice from which the measurements were pooled (black), and the value of the varied immunization scheme parameter, corresponding to dosage $D$ (pink) in μg of TT for the first two schemes and time delay $\Delta T$ (blue) in days for the third.

progressively maximize the likelihood $\mathcal{L}$ for the selection variant (A), (B), and (C). For each point in the parameter space, the deterministic model is simulated according to the immunization scheme considered, see *Figure 5*. In particular, for scheme one the prediction consists in the simulation of a single GC with variable injected dosage $D$ lasting at most for 4 weeks. Since cells are harvested 4 days after the second injection, we consider the Ab-SCs population to be comprised of a mixture of MC and PC according to the MC fraction $g_{\mathrm{imm}}$, whose value is inferred to be zero (i.e. in this case the Ab-SC population comprises only PCs). For scheme 2 and 3 instead we consider two GC simulations, one per injection. We vary either the injected dosage $D$ or the time between the two injections $\Delta T$ according to the protocol in exam. The second GC simulation, initiated 6 days after the second

injection, can be seeded by MCs collected during the first GC evolution up to the injection time. Moreover, since cells are harvested 1 day after boost we consider the MC fraction in the Ab-SC population to be $g_{\mathrm{recall}}$. The affinity distribution of Ab-SCs obtained with the deterministic model is then used to compute the likelihood of the experimentally measured affinities of the IgG-SCs, under all tested immunization schemes. Cells sampled from the spleen can originate from different GCs, but, as long as these GCs have equal defining parameters, their average evolution is the same, and their multiplicity does not affect the inference procedure. See Materials and methods and appendix sect. 4 ('Maximum likelihood fit procedure') for a more detailed description of the procedure. Notice that the inference of many parameters is made possible by the richness of information contained in the experimental affinity distributions.

As an outcome, we obtain the log-likelihoods of the three variants listed above: $\ln \mathcal{L}^{(A)} = -7400.37$ for full two-step selection, $\ln \mathcal{L}^{(B)} = -7459.39$ for non-permissive two-step selection, and $\ln \mathcal{L}^{(C)} = -7400.67$ for T-cell-based selection (see *Appendix 1—figure 9* for the inferred parameters value in all cases). A fair comparison between these three hypothesis must however acknowledge that (B) and (C) have, respectively, 2 and 1 less parameter to fit the data than (A). We therefore resort to the so-called Bayesian Information Criterion (BIC), which takes into account the number of parameters by estimating the volume in the parameter space around the peak in likelihood. BIC is defined as $k \ln n - 2 \ln \mathcal{L}$, where $k$ is the number of parameters in the model and $n$ is the number of data points available for the inference. We obtain BIC$^{(A)}$ = 14877.3, BIC$^{(B)}$ = 14978.3, BIC$^{(C)}$ = 14869.4. We conclude that the model to be chosen (with lowest BIC) is (C) (Notice that variant (C) is also preferred based on an alternative to BIC, the Akaike Information Criterion, defined through AIC = $2k - 2 \ln \mathcal{L}$ (AIC$^{(A)}$ = 14818.7, AIC$^{(B)}$ = 14932.8, AIC$^{(C)}$ = 14817.3). Including Ag-binding selection improves slightly the likelihood, but less than expected from the introduction of an extra parameter ($\epsilon_{Ag}$). On the contrary, the large increase in BIC when forbidding permissiveness shows that non-zero values for $a, b$ are definitely needed to fit the data. Within variant (C) the values of the eight model parameters that maximize the likelihood are (see *Table 1*): $k_B^- = 2.07 \times 10^{-5}\,/\mathrm{d}$, $\mu_{\mathrm{naive}} = -14.59$, $\sigma_{\mathrm{naive}} = 1.66$, $a = .12$, $b = .66$, as well as $g_{\mathrm{recall}} = 56\%, g_{\mathrm{imm}} = 0\%$ for the values of $g$ corresponding to the measurements of affinities, respectively, 1 day after boost injection (immunization scheme 2 and 3) or 4 days after the second injection (scheme 1), and the dosage-to-concentration conversion factor $\alpha = 23\,\mathrm{ng}$ which allows us to convert Ag dosages in units of mass into dimensionless concentrations. In *Figure 5* we report for every experimentally measured affinity distribution (orange histograms) the maximum-likelihood corresponding prediction according to the deterministic model evolution (blue curves; for good comparison normalization considers only the area of the curve below the experimental sensitivity threshold). Under all tested immunization schemes, we observe a very good agreement between theory and experiments. See *Appendix 1—figure 7* for the full plot including all experimental conditions.

## Effect of varying Ag dosage and time between injections

In *Figure 6*, we report average measures performed on the affinity distributions for the three different schemes (scheme 1 to 3, left to right) considered. The measurements are the average binding affinity (top) and the high energy fraction (bottom). The latter is defined as the fraction of cells in the population having binding affinity higher than $K_d^{\mathrm{h-aff}} = 50\,\mathrm{nM}$, or equivalently $\epsilon < \epsilon_{\mathrm{h-aff}} = -16.8$. In the figure we compare experimental values (orange) with the theoretical prediction of the deterministic model (blue line) and the stochastic simulations (light green shaded area corresponds to the standard deviation over 1000 stochastic simulations). To convey a measure of experimental individual variability, for each immunization scheme we report single-mouse measurements as orange crosses, connected by vertical lines; Orange empty dots represent instead averages over the pooled data. As cells measured from a mouse spleen can originate from different GCs, for example 20 to 50 GCs per spleen section were reported in *Wittenbrink et al., 2011*, we also display in a darker shade of green the standard deviation of the mean of 20 simulations of the stochastic model. This allows us to estimate the expected variations of the binding energy or other quantities due to the existence of multiple GCs. For all the schemes considered, we observe a very good agreement between the stochastic model and theoretical predictions, showing that the infinite size limit is a good approximation to the average stochastic evolution. This agreement also extends to full distributions (compare green histograms and blue curves in *Appendix 1—figure 7*).

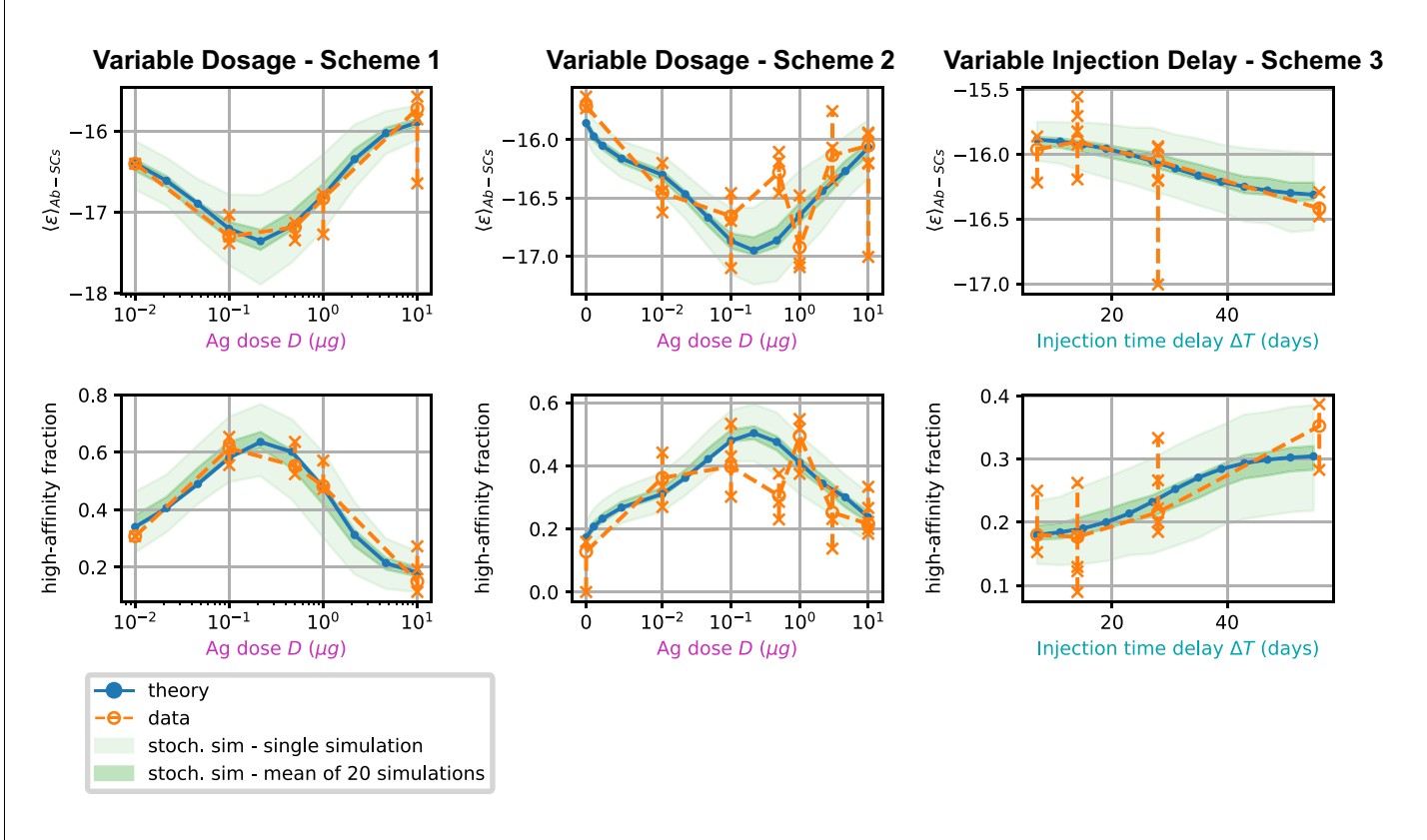

**Figure 6.** Comparison between data and model prediction for the average binding energy (top) and high affinity fraction (bottom) of the Ab-secreting cell population under the three different immunization schemes (scheme 1 - left, scheme 2 - center, scheme 3 - right). The high-affinity fraction corresponds to the fraction of measured cells having binding affinity $K_d < 50\,\mathrm{nM}$, or equivalently binding energy $\epsilon < -16.8\,k_BT$. On the x axis we report the variable quantity in the scheme, which is administered dosage $D$ for schemes 1 and 2 and delay between injection $\Delta T$ for scheme 3. Green shaded areas indicate the results of the stochastic model simulations. The light area covers one standard deviation around the average result for a single simulation, while the dark area corresponds to the standard deviation for the mean over 20 simulations. This quantifies the expected variation for populations of cells extracted from a spleen, that could potentially have been generated by many different GCs. Results are evaluated over 1000 different stochastic simulations per condition tested. The deterministic solution of the model, in blue, reproduces well the average over stochastic simulations in all the considered schemes. Data coming from experimental affinity measurement of IgG-secreting cells extracted from spleen of immunized mice are reported in orange. Orange empty dots represent averages over the data pooled from multiple mice immunized according to the same scheme, while orange crosses represent averages for measurements from a single mice. Crosses are connected with a vertical dashed line in order to convey a measure of individual variability. Notice that the number of mice per scheme considered can vary, see *Figure 5* and *Appendix 1— figure 7*). In order to compare these data with our model, both for the stochastic simulations and the theoretical solution we take into account the experimental sensitivity range when evaluating averages.

Most importantly, for all schemes, model and data are in very good agreement. In schemes 1 and 2 in particular both show the existence of an optimal intermediate dosage corresponding to maximal affinity of the Ab-secreting cells recalled population. This experimental observation can now be interpreted with the theoretical analysis introduced in section (Theoretical analysis at fixed concentration).

In scheme 3, we observe that experimental data show a slight increase in affinity for longer injection delays, and so does our model. This is presumably due to a combination of two effects. Firstly, the fact that higher affinity cells are produced late in the response, and waiting more before harvesting cells allows for higher affinity cells to be created. Secondly, giving the first GC time to produce high-affinity MCs is beneficial since then these cells can then colonize the second GC and continue their maturation even further there.

# Discussion

## Summary and significance

In this paper, we have investigated the relationship between Ag dosage and quality of immunization outcome. Several studies (*Victora and Nussenzweig, 2012*; *Kang et al., 2015*; *Eisen and Siskind, 1964*; *Goidl et al., 1968*; *Nussenzweig and Benacerraf, 1967*) report the fact that better affinity maturation is not always favored by higher doses of Ag, but can instead be enhanced by lower doses. Similarly, the strength of a response to a vaccine, usually measured through the count of responding cells, may show a bell-like curve at intermediate dosages, and understanding the mechanisms underlying this behavior and locating the optimal Ag dose are of crucial importance (*Rhodes et al., 2019*). Our works provide quantitative theoretical and experimental support to these findings. In particular, the stochastic model for affinity maturation we consider here is capable of explaining and accounting for the existence of an intermediate optimal Ag dosage, that results in the highest average affinity of the recalled population. While our model is inspired by previous studies of the evolution of a population of B-cells in a Germinal Center during Affinity Maturation, such as (*Wang et al., 2015*, it differs in two substantial ways.

First, our model is amenable to detailed mathematical analysis. We show that the stochastic evolution of the distribution of binding energies can be accurately approximated by a deterministic dynamics (see *Figure 6*), which we resolve exactly. Under constant Ag concentration, the distribution of binding energies behaves as a traveling wave, whose speed and growth rate can be recovered by solving an appropriate eigenvalue *Equation 4*. The dependence of these two quantities on Ag concentration reveals the role Ag availability plays in controlling the strength of selection, both in the generated data and models. In particular, high Ag dosage results in low selection pressure and no maturation, and conversely too low Ag dosage in high selection pressure and population extinction. Only intermediate Ag concentration and intermediate selection pressure ensures both population survival and successful AM.

Second, we show that a single set of parameters of our model is able to reproduce quantitatively the many distributions of single-cell affinities measured on IgG-SC extracted from mice immunized against TT corresponding to multiple protocols largely varying in Ag dosages and delays between injections. To determine the best parameters, we introduce a maximum-likelihood-based inference method. Our inference method fully exploits the results of the experimental technique, developed in *Eyer et al., 2017*, giving access not to the average affinity, as titer measurement would, but to the complete affinity distribution of the recalled Ab-SC population. This population information is crucial for accurate inference of the model parameters and for a meaningful validation of the model. Furthermore, the inferred parameters provide insights on the internal processes of affinity maturation, such as on the role of permissiveness, as discussed later. Inference techniques are powerful instruments in this respect, since they help us investigate experimentally unaccessible features of the system through their indirect but measurable effects. Our inference procedure is very flexible and can readily be applied to new datasets, providing ad-hoc estimates of parameters for different antigens or even different organisms.

## Maturation as combination of beneficial mutations and selection of high-affinity precursors

Our stochastic model for affinity maturation is subject to homogenizing selection (*Figure 3*), to degrees depending on the presence of a high-affinity precursor (*Figure 3D*), in agreement with experimental evidence (*Tas et al., 2016*; *Abbott et al., 2018*). In addition, the initial choice of founder clones accounts for a large part of the stochasticity in the maturation outcome (*Appendix 1—figure 3E to H*), Hence, in our model, selective expansion of high-affinity precursors plays an important role in affinity enhancement. Affinity enhancement is also obtained through the accumulation of beneficial mutations. When observing the distribution of beneficial and deleterious mutations in the MC and PC populations (*Appendix 1—figure 10*), one finds that, even though on average cells accumulate very few mutations during the AM process, selection tends to favor the fixation of beneficial mutations and the disappearance of deleterious ones.

## Stochastic effects in fitness waves

Both the mathematical analysis and the inference procedure are made possible by the fact that our stochastic maturation model is well-approximated by its deterministic counterpart. This is usually not the case when describing the evolution of fitness waves (*Neher and Walczak, 2018*). In many systems, stochastic fluctuations may play a major role, for example when the evolving population passes through a bottleneck, and transiently has very low size, before increasing again. Fluctuations may also be acquire crucial importance when the evolution lasts so long that the leading edge of the fitness wave has time to exponentially amplify and govern the bulk of the population. Here, experimentally measured quantities, such as the distribution of affinities, are the outcome of an average over multiple GC reactions in the spleen. While single simulated GCs show signs of individuality, see homogenizing selection and the evolution of clonality in *Figure 3*, the average product of multiple GCs is well-approximated by our deterministic theory. Moreover, stochastic effects are also partially mitigated by the fact that we consider quantities related to the integral over time of the fitness wave evolution, namely, the MC and PC distributions. Hence temporal fluctuations are smoothed out. Another factor contributing to this mitigation is the limited selection we infer. The permissiveness of selection results in a less drastic decrease of the population size, and a reduced sensitivity to fluctuations from the leading edge of the fitness wave.

## Permissiveness in GC selection

The role of permissiveness in germinal center selection is still an open question (*Bannard and Cyster, 2017*; *Mesin et al., 2016*; *Victora and Mouquet, 2018*). Through phenomena such as bystander activation (*Horns et al., 2019*) and stochastic noise, GC selection may also allow intermediate- and low-affinity clones to survive, rather than maturing exclusively via selection of the few best clones (*Lau and Brink, 2020*; *Tas et al., 2016*). These phenomena generate a wider diversity than previously appreciated, especially when considering complex Ags displaying different epitopes (*Kuraoka et al., 2016*). In *Finney et al., 2018* for example the authors try to characterize the GC response to complex Ags such as influenza vaccine, as opposed to simple ones such as haptens. While in the latter case a strong homogenizing selection and affinity maturation is observed, for complex Ags response is more polyclonal and a consistent part of the GC population (20–30%) is composed of low-affinity clones. This suggests a more permissive nature of the GC selection, in which even low-affinity clones have a non-zero probability of receiving T-cell help.

  To model these effects, we have introduced two parameters, $a$ and $b$, in the competitive selection process involving survival signals from $T$-helper cells, see *Equation 2* and *Appendix 1—figure 1D*. $a$ corresponds to the baseline probability for cells to survive a selection step, while $b$ is equal to the probability for cells to fail selection even if they have high affinity; this could be due for example to the limited availability of T-cell help, which could increase the stochasticity of the selection process (*Krishna and Bachman, 2018*). The role of the parameters $a$ and $b$ in controlling the population evolution is studied in section appendix (Permissive and stochastic selection: effect of a, b parameters Our maximum likelihood fit of the data yields $a = 0.12$ and $b = 0.66$. These values imply that the probability that a high-affinity cell to survive the second step of selection is $1 - b = 34\%$, about two and a half times the probability for a low-affinity cell, given by $a = 12\%$. This observation is in support for the permissive and stochastic nature of selection, at least in our experimental conditions. The non-permissive variant of our model with base-line levels $a = b = 0$ in *Equation 2*, referred to as variant (B), offers a much worse fit of the data, even when taking into account the smaller number of parameters of this variant (see appendix 'Possible model variations').

## GCs entry selection does not seem to be restrictive

Our inference procedure supports the statistical prevalence of variant (C), with T-cell-based selection only, with respect to (A), which included Ag-binding selection. The fact that Ag-binding selection does not seem to be a limiting step for GC colonization, at least in the range of our experimentally measurable affinities, is compatible with experiments performed in *Schwickert et al., 2011*, in which it is shown that in absence of high affinity competitors even clones with low affinity (as low as $K_d \sim 8\,\mu\mathrm{M}$ or equivalently $\epsilon \sim -11.7$) can colonize GCs. This is also in accordance with the fact that selection in GCs should be relatively permissive (*Bannard and Cyster, 2017*; *Victora and Nussenzweig, 2012*) in order not to limit the diversity of the repertoire. Let us emphasize, however, that the

difference in the BIC of the two selection models is rather weak and that our conclusion is contingent on the data set collected and analyzed here.

## Fractions of PCs and MCs amongst Ab-SCs

Our experimental setup does not allow us to identify whether the IgG-SCs we observe originate from reactivated MCs or residual PCs generated during previous immunizations. We therefore compared the experimental measurements with a weighted mixture of the MC and PC populations predicted by our model. This mixture, which we call the Ab-SC population, represents the population of cells that respond to antigenic challenge under particular conditions. We introduced the parameters $g_{recall}$ and $g_{imm}$, corresponding to the fraction of reactivated MCs in the Ab-SC populations when measurement is performed one day after boost or four days after the second injection, and fit their value on the experimental measurements. The result of our inference procedure indicates that, when the system is probed 1 day after pure TT boost, most of the response consists in reactivated memory cells ($g_{recall} = 56\%$). This is in agreement with experimental observations performed in *Eyer et al., 2020*, in which the frequency of IgG-SCs increased from $0.6 \pm 0.1\%$ to $1.6 \pm 0.2\%$ one day after the boost, indicating that around 64% of IgG-SCs were not present before the boost. When the measurement is performed 4 days after the second injection then we predict that the vast majority of responders consist of residual PCs ($g_{imm} = 0\%$, with a confidence interval extending to 6%, *Appendix 1—figure 6*). This is consistent with experimental data (unpublished), which indicate that the majority of IgG-SCs are still active 28 days after CFA immunization, and will be secreting at +4 days.

Concerning the biological difference between the MC and PC populations, it has been observed that MCs show on average less maturation than PCs (*Inoue et al., 2018*; *Shinnakasu et al., 2016*; *Shinnakasu and Kurosaki, 2017*), a feature that is reproduced in our model (*Figure 2D*) as a consequence of the temporal switch we introduced (*Appendix 1—figure 1B*) and might be important in maintaining diversity in the response, especially against highly mutable pathogens, and mitigating original antigenic sin (*Suan et al., 2017*; *Morens et al., 2010*). The results of our inference are in agreement with the fact that experimentally we observe a higher affinity of the responders if measurement is performed 4 days after the last injection (scheme 1) rather than 1 day after boost (schemes 2,3). This difference in affinity could also originate from some form of selection acting on the responder population during the first days of the response, which could selectively expand high-affinity clones in the time between Ag challenge and measurement. Including this selection in the model would result in a different estimate of the fractions $g_{recall}$ and $g_{imm}$. However, for simplicity and lack of explicit experimental evidence we did not include this selection in the model.

## Model limitations and discussion

In building the model, we chose to only keep the minimal features that could allow us to understand the existence of an optimal dosage and be able to reproduce experimental observations, while still being mathematically tractable. Among the simplifications, the number of duplications per cell is considered independent of the cell affinity. It has been, however, shown that an affinity correlates with GC dark zone dwelling time and number of divisions (*Gitlin et al., 2015*). This phenomenon introduces an effective fitness difference, which is in practice qualitatively accounted for by the selection terms in our model. Moreover, we consider the distribution of affinity-affecting mutations $K_{aa}(\Delta\epsilon)$ to be independent of the clone's affinity, similarly to *Wang et al., 2015*; *Wang, 2017*; *Zhang and Shakhnovich, 2010*. In reality, independence holds only away from affinity peaks in the Ab sequence space; close to these peaks, affinity-increasing mutations become rare, and it is expected that Ag affinity of clones eventually saturate, while the binding energy can take arbitrarily low values in our model. However, in the regime defined by the values of the parameters inferred on our experimental data, MCs and PCs generated by our stochastic model accumulate on average very few mutations in the course of evolution (appendix, Quantifying beneficial and deleterious mutation events and *Appendix 1—figure 10*), with the maximum number of beneficial mutations accumulated being compatible with experimental evidence (*Gérard et al., 2020*) ( appendix Quantifying beneficial and deleterious mutation events). In this regime, mutations account for only a part of the maturation, the rest being achieved through selection of high-affinity founder clones ( appendix Quantifying beneficial and deleterious mutation events). This is in line with the limited maturation

observed in our experiments. In cases where the saturation effect may become relevant, other approaches to model the effect of affinity-affecting mutations might be more appropriate, for example the introduction of a 'shape space' representation (*Shaffer et al., 2016*; *Wang et al., 2016*). The model and results reported here do not include Ab-feedback (*Wang et al., 2015*), the phenomenon by which GC B-cells not only have to compete amongst themselves for Ag acquisition but also with Abs produced earlier in maturation (*Bannard and Cyster, 2017*; *Mesin et al., 2016*; *Zhang et al., 2013*), which could prevent B-cells from internalizing Ag by binding to it. We did not include Ab-feedback in our model, however preliminary investigations (not shown) suggest that it would not affect the existence of an optimal dosage range. GC lifetimes reported in literature vary considerably, from 1 to 2 weeks for soluble protein boosting to several months or longer for certain infections (*Victora and Mouquet, 2018*; *Mesin et al., 2016*). In alum immunizations GC lifetimes of 3–4 weeks have been observed (*Takahashi et al., 1998*). In our simulations, a long lifetime for GCs is observed and for a high dose of Ag they can have an effective lifetime lasting up to 3 months (*Appendix 1—figure 2E*). The concentration of Ag is crucial in determining the strength of selection and the lifetime of the GC in our model. In reality, Ag dosage value also controls the initiation of the GC and AM. In particular, one could expect that for very low dosages the GC reaction would not be initiated at all. For simplicity, we avoid including this phenomenon in our model, and GC reaction takes place in our simulations even at very low Ag dosages, with the result that very few, highly affine MC are produced in this regime. To avoid a discontinuity with respect to the case of null Ag dosage, $D = 0\,\mu g$, in which we expect the measured B-cell population to originate directly from naive precursors, we perform differentiation at the beginning of the simulation round, before mutations and selection (see appendix 'Model definition and parameters choice'). This generates a core of low-affinity MCs keeping the average affinity of the population close to $\mu_{\mathrm{naive}}$, even when few additional high-affinity MCs are added. However, this might be an unnecessary caution, since when looking at the data we observe that even the lowest tested dosage ($D = 0.01\,\mu g$ TT, *Figure 5*) shows the hallmark of maturation when compared to the the case of zero dosage ($D = 0\,\mu g$ TT, *Figure 5*). This signals that in the dosage range considered in our experiments we expect maturation to occur. Furthermore, in our model Ag inputs, for example resulting from a new injection, cannot enter a GC while the maturation process is ongoing. Our choice is partly justified by the observation that injecting an Ag bolus when a GC maturation process is in place mostly results in disruption of the ongoing GC reaction (*Victora and Nussenzweig, 2012*; *Pulendran et al., 1995*; *Shokat and Goodnow, 1995*; *Han et al., 1995*; *Victora et al., 2010*). We only model a single 'average' GC, whose output is assumed to be representative of the outcome of AM. In reality, MC and PC populations are generated by many parallel GC reactions, which could in principle weakly interact via invasion of clones from one GC to another (*Mesin et al., 2016*; *Victora and Mouquet, 2018*). Last of all, to test the robustness of some of our hypothesis we performed the inference procedure under slightly different conditions. In particuar, we considered the effect of increasing the Ag decay rate, of setting $p_{\mathrm{mem}}$ to be a constant and not depend on the number of MCs accumulated during evolution, and also of considering the MC/PC time-switch to be only partial, with a residual production of MCs all along the evolution. We verified that even in these case the model is in good agreement with the data. The results are reported in appendix sect. 6.

## Outlooks

As shown above our model for AM is simple enough to be amenable to detailed mathematical analysis and, yet, is able to accurately reproduce the full affinity distributions of Ab-SCs generated during the immunization process. This finding suggests several extensions to the current work. First our model could be used to predict the outcome of more complex immunization protocols than the ones investigated experimentally in this work. In particular, it would be interesting to consider the case of continuous delivery methods (osmotic pumps, repeated injections...) (*Tam et al., 2016*; *Cirelli et al., 2019*), through which the Ag concentration can be precisely controlled over time, and make predictions for the optimal delivery process. Secondly, the quantitative fit of the model parameters was made here possible thanks to the maximum-likelihood algorithm we have introduced, which is flexible and robust. Our inference procedure, whose code is made available with the publication (see Materials and methods code and data availability), could be readily applied to to different measurements, as well as to variants of the present models, with extra parameters corresponding to features of the affinity maturation process that are hardly experimentally

accessible, such as selection permissiveness. The combination of quantitative modeling with inference appears as a promising tool to understand the mechanisms governing the immune response and to guide the development of strategies to control and direct it.

## Materials and methods

### Experimental procedure

#### Observation chamber assembly

For the 2D observation chamber, we used glass microscopy slides as top and bottom covers (76 × 26×1 mm, Marienfeld). Two access holes of 1 mm diameter were generated in the top glass slide using laser ablation (C180II, Axys Laser). Afterwards, both slides were thoroughly cleaned using soap, water and ethanol, and the two glass slides were exposed to air plasma (60 W) for 10 min (Femto, Diener Electronics). After plasma treatment, double sided thermos-responsive tape (series 1375, Orafol), beforehand cut into shape using a cutting plotter (CE-6000–40, Graphtec), was stuck onto the glass slides and the chamber sealed. The chamber was heated to 150° C and pressed with 7 bar for 5 min to reduce the height to enable a monolayer of droplets only. Next, two nanoports (N333-01, Idex) were glued to the access holes. Subsequently, the surface of the 2D chamber was treated using fluoro-silane (Aquapel, Aquapel) to render the surface hydrophobic. Lastly, the chamber was dried under nitrogen, and subsequently filled with fluorinated oil (Novec HFE7500, 3M) and sealed until used. The chamber was re-used multiple times, and when properly stored, was used for up to 2 months. Cleaning was performed after each experiment by flushing fluorinated oil to remove droplets, and the chamber was stored filled with HFE7500 until the next use.

#### Droplet generator

Microfluidic PDMS chip for droplet generation were fabricated as previously described (*Eyer et al., 2017*).

#### Aqueous phase I

Preparation of cells for droplet creation. For droplet generation, cellular suspensions were centrifuged (300 g, 5 min). and washed once in droplet media comprising RPMI w/o phenol red with supplemented 5% KnockOut Serum Replacement (both ThermoFisher), 0.5% recombinant human serum albumin (A9986, Sigma), 25 mM HEPES pH 7.4, 1% Pen/Strep and 0.1% Pluronic F-137 (all ThermoFisher). The cells were re-suspended in droplet media to achieve a $\lambda$ (mean number of cells per droplet) of 0.2–0.4.

#### Aqueous phase II

Beads and reagents. Paramagnetic nanoparticles were prepared as described before (*Eyer et al., 2020*). Before use, the nanoparticles were re-suspended thoroughly.

#### Data acquisition

Droplets were generated as previously described (*Eyer et al., 2017*), and the emulsion was directly injected into the 2D observation chamber. After chamber filling was complete, the chamber was gently closed and mounted onto an inverted fluorescence microscope (Ti Eclipse, Nikon). Two neodymium magnets (BZX082, K and J Magnetics) were placed on each side of the chamber during observation to hold the bead lines in place. Excitation light was provided by a LED source (SOLA light engine, Lumencor Inc). Fluorescence for the specific channels were recorded using appropriate band pass filters (GFP and TRITC filter sets, Nikon, and Cy5 filter set, Semrock) and camera settings (Orca Flash 4, Hamamatsu) at room temperature (25° C) and ambient oxygen concentration. Images were acquired using a 10x objective (NA 0.45). An array of 10 × 10 images were acquired for each experiment, every 7.5 min in all channels over 37.5 min (five measurements total).

#### Data analysis

Data was analysed using a custom-made Matlab script (Mathworks). The resulting raw data were exported to Excel (Microsoft), and sorted for droplets that showed an increase in anti-IgG relocation

over time above a threshold (*Eyer et al., 2017*). The selected droplets were controlled visually for the presence of a cell, and the absence of any fluorescent particles, relocation on cells (i.e. dead cells) or droplet movement. The so-selected droplets were analyzed to calculate dissociation constants as described previously (*Eyer et al., 2017*). The limit of detection of the instrument allows for the resolution of dissociation constants $K_d \leq 500\,\mathrm{nM}$ (see *Eyer et al., 2017*), therefore measurements with lower affinity were discarded. Moreover, cells with very high affinity $K_d < 0.1\,\mathrm{nM}$ could be observed, but their affinity could not be determined more precisely (see also *Eyer et al., 2017*) and was set to $K_d = 0.1\,\mathrm{nM}$.

## Immunization of mice

The mice used herein were part of the study as published elsewhere (*Eyer et al., 2020*). In short, BALB-C mice were purchased from Janvier Labs (age 6–8 weeks at start, female) and housed in the animal facilities of Institute Pasteur during experimentation. All immunizations were made intraperitoneal. Each condition was replicated three times in the same cohort ($n = 3, N = 1$); except when explicitly stated otherwise. From each mouse, between $20' - 100'000$ cells were assayed in an experimental run.

## Extraction of IgG-SCs

Spleens were harvested at the indicated time points of the immunization schedule. Spleen cell suspensions were recovered following disassociating using a 40 µm cell strainer. Cellular suspensions were pelleted at 300 g for 5 min, and red blood cell lysis was performed for 1 min using BD Pharm Lyse (BD). Cells were washed twice with MACS buffer and re-suspended in 3 ml of MACS buffer. These cells were further processed according to the manufacturer's protocol using the Pan B Cell Isolation Kit II (Miltenyi) on a MultiMACS Cell24 Separator Plus (Miltenyi, program depletion). Purity of B-cell lineage was usually above 90% (data not shown).

## Antigen dynamics

Ag concentration dynamically changes in parallel with the evolution of the GC. Initially an amount $C_{\mathrm{inj}}$ of Ag is administered through injection (*Figure 2A*). The value of $C_{\mathrm{inj}}$ determines the initial amount of Ag trapped in the adjuvant matrix, setting the initial value of the Ag reservoir concentration (*Equation 5*, right). For the sake of comparison with experiments $C_{\mathrm{inj}}$ is proportional to the injected Ag dosage $D$ up to a conversion factor $\alpha$, $C_{\mathrm{inj}} = D/\alpha$, inferred through maximum likelihood fit of the data. The available ($C_{\mathrm{av}}$, appearing in the selection probabilities in *Equations 1 and 2*) and reservoir ($C_{\mathrm{res}}$) concentrations then evolve as described in the main text (*Figure 2A*) under the action of release, decay and consumption according to the following equations:

$$\frac{d}{dt}C_{\mathrm{res}}(t) = -k^{+}C_{\mathrm{res}}(t), \quad C_{\mathrm{res}}(t=0) = C_{\mathrm{inj}} \tag{5}$$

$$\frac{d}{dt}C_{\mathrm{av}}(t) = k^{+}C_{\mathrm{res}}(t) - (k_{\emptyset}^{-} + k_{B}^{-}N_{t}^{B})\,C_{\mathrm{av}}(t) \tag{6}$$

During GC formation ($t < T_{\mathrm{GCformation}} = 6\,\mathrm{d}$) the number of B-cells, $N_{t}^{B}$, appearing in the rate of Ag consumption increases exponentially up to the maximal size $N_{i}^{B} = 2,500$. More details on the concentration evolution can be found in appendix Model definition and parameters choice.

## Deterministic evolution

In the deterministic/infinite size approximation, the stochastic processes that model one round of GC maturation can be written as operators acting on the distribution $\rho$ of binding energies $\epsilon$.

- Cell duplication is represented by the amplification operator, consisting in a simple multiplication:

$$\mathbf{A}\,[\rho](\epsilon) = 2\,\rho(\epsilon) \tag{7}$$

- Mutations are encoded by convolution of the distribution of energies with a mutation kernel $K_{\mathrm{eff}}$ that includes the effect of silent, affinity affecting and lethal mutations (see *Equation 26* and appendix sect. 3 'Theoretical solution and eigenvalue equation'):

$$\mathbf{M}\left[\rho\right](\epsilon) = \int d\Delta\epsilon \, K_{\mathrm{eff}}(\Delta\epsilon) \, \rho(\epsilon - \Delta\epsilon) \tag{8}$$

- Selection for Ag binding (*Equation 9*) and T-cell help (*Equation 10*) are encoded simply by a product with the respective probabilities (*Equations 1 and 2*), where for the latter the probability depends on $\bar{\epsilon}$ which in turns depends on the distribution of binding energies, making the operator not linear:

$$\mathbf{S}_{\mathrm{Ag}}\left[\rho\right](\epsilon) = P_{\mathrm{Ag}}(\epsilon) \times \rho(\epsilon) \tag{9}$$

$$\mathbf{S}_{\mathrm{T}}\left[\rho\right](\epsilon) = P_{\mathrm{T}}(\epsilon, \bar{\epsilon}) \times \rho(\epsilon), \quad \text{with} \quad e^{-\bar{\epsilon}} = \frac{\int d\epsilon \, \rho(\epsilon) \, e^{-\epsilon}}{\int d\epsilon \, \rho(\epsilon)}. \tag{10}$$

- Finally, the carrying capacity (*Equation 11*) and the differentiation (*Equation 12*) processes correspond to multiplications:

$$\mathbf{N}\left[\rho\right](\epsilon) = \min\left\{1, N_{\max}^{B}/N^{B}\right\} \times \rho(\epsilon), \quad \text{with} \quad N^{B} = \int d\epsilon \, \rho(\epsilon), \tag{11}$$

$$\mathbf{D}\left[\rho\right](\epsilon) = (1 - p_{\mathrm{diff}}) \times \rho(\epsilon) \tag{12}$$

The distribution of binding energies at round $t$ then evolves through $\rho_{t+1} = \mathbf{E}\left[\rho_t\right]$, where the complete operator is $\mathbf{E} = \mathbf{D}\mathbf{N}\mathbf{S}_{\mathrm{T}}\mathbf{S}_{\mathrm{Ag}}\mathbf{R}$, we indicate with $\mathbf{R} = \mathbf{M}\mathbf{A}\mathbf{M}\mathbf{A}$ the operator encoding for two rounds of mutations and amplification. The evolution operator features, in order of application, two rounds of amplification and mutation, Ag-binding selection, T-cell help selection, carrying capacity and differentiation. Notice that for variant (C), there is no Ag-binding selection, and $\mathbf{S}_{\mathrm{Ag}}$ is replaced with the identity operator.

## Eigenvalue equation and phase diagram

The growth rate $\phi$ and the maturation velocity $u$ shown in *Figure 4* are characteristic of the travelling wave nature of the distribution of energies $\rho$ at large 'times'. When Ag-binding selection is irrelevant (as is the case at large times if $u<0$) and the carrying capacity constraint is omitted, the evolution operator simplifies into

$$\mathbf{E} = \mathbf{D}\mathbf{N}\mathbf{S}_{\mathrm{T}}\mathbf{S}_{\mathrm{Ag}}\mathbf{R} \rightarrow \mathbf{D}\mathbf{S}_{\mathrm{T}}\mathbf{R} \tag{13}$$

In one round of maturation, we expect the travelling distribution $\rho^*(\epsilon)$ to be shifted by $u$ along the energy axis, and to be multiplied by $e^{\phi}$. Without loss of generality, we may choose $\rho^*$ such that $\bar{\epsilon} = 0$; any other choice would merely consists in a translation of $\rho^*$ along the energy axis. Hence, $\mathbf{E}$ is now a linear operator, and $\rho^*$ satisfies the eigenvalue *Equation 4*. In other words, the operator $\Sigma(-u) \cdot \mathbf{E}$, where $\Sigma(-u)$ is the shift operator $\epsilon \rightarrow \epsilon + u$, has for largest eigenvalue $e^{\phi}$ and associated eigenvector $\rho^*$. As all the entries of $\Sigma(-u) \cdot \mathbf{E}$ are positive, the Perron-Frobenius theorem ensures that $e^{\phi}$ is the top eigenvalue associated to the unique eigenvector $\rho^*$ with all its components positive.

In practice, given a guess value for $u$, one can iterate $\Sigma(-u) \cdot \mathbf{E}$ a sufficient number of times to determine its top eigenvector $v(\epsilon; u)$, and compute $\bar{\epsilon}(u)$ through

$$e^{-\bar{\epsilon}(u)} = \frac{\int d\epsilon \, e^{-\epsilon} \, \mathbf{R} \, v(\epsilon; u)}{\int d\epsilon \, v(\epsilon; u)}. \tag{14}$$

The value of $u$ is then tuned until $\bar{\epsilon}(u) = 0$. For a graphical representation of the resolution procedure and details on the numerical scheme used, see *Appendix 1—figure 4* and appendix Theoretical solution and eigenvalue equation.

## Maximum likelihood parameters determination

Nine parameters of the model have been obtained through maximum likelihood fit of the data:

- the conversion factor $\alpha$, which allows for conversion between experimental administered Ag dosage $D$, measured in micrograms, and the dimensionless administered Ag concentration of our model, $C = D/\alpha$.
- the Ag consumption rate per B-cell $k_B^-$, which controls the GC lifetime and also the extent of the affinity maturation.
- the mean $\mu_{\text{naive}}$ and variance $\sigma^2_{\text{naive}}$ of the Gaussian binding energy distribution for the GC seeder clones, elicited directly from the naive population.
- the binding energy threshold $\epsilon_{\text{Ag}}$ for a B-cell to be able to bind Ag with sufficient affinity to internalize it (cf *Equation 1*). This parameter does not appear in variant (C), where selection is mediated by T-helper cells only.
- The T-cell selection characteristic coefficients, $a$ and $b$, encoding respectively the baseline probabilities to survive or not survive selection, see *Equation 2* and *Appendix 1—figure 1.D*.
- The weight parameters $g_{\text{recall}}$, $g_{\text{imm}}$, representing the MC fraction in the measured population of IgG-SCs for the two protocols, respectively for schemes 2 and 3 with measurement one day after boost, and scheme 1 with measurement 4 days after second injection.

We use a procedure that maximizes the average likelihood of experimental affinity measurements. To do so, we perform the following steps:

1. For each of the 15 different experimental conditions $\mathcal{S}$ (five different dosages in scheme 1, plus seven different dosages in scheme 2, plus four different injection delays in scheme 3, minus the experiment at dosage 10 $\mu$g TT and 4 weeks delay between injection which is repeated, being present in both schemes 2 and 3) we evaluate the log-likelihood of the experimental measurements through

$$\ln \mathcal{L}(\mathcal{S}) = \sum_{s \in \mathcal{S}} \ln \rho_{\text{Ab-SC}}(\epsilon_s, \mathcal{S}) \tag{15}$$

2. where $\{\epsilon_s\}_{s \in \mathcal{S}}$ are the binding energy ($\log K_D$) single-cell measurements performed in condition $\mathcal{S}$, and $\rho_{\text{Ab-SC}}(\epsilon_s, \mathcal{S})$ is the normalized distribution of binding energies of Ab-SC predicted by the deterministic version of the model, and defined as a weighted sum of the normalized MC and PC distributions with MC fraction $g$: $\rho_{\text{Ab-SC}}(\epsilon_s, \mathcal{S}) = g \, \rho_{\text{MC}}(\epsilon_s, \mathcal{S}) + (1 - g) \, \rho_{\text{PC}}(\epsilon_s, \mathcal{S})$. This fraction is either $g_{\text{recall}}$ or $g_{\text{imm}}$, depending on the condition $\mathcal{S}$ considered (scheme 2,3 or scheme 1). For good comparison with the data the final normalization is done for the part of the distribution inside the experimental sensitivity range $-23.03 = \epsilon_{\min} < \epsilon < \epsilon_{\max} = -14.51$. Notice that a measurement equal to $\epsilon_{\min}$ could in truth originate from any lower value of the energy, a situation not taken into account in the above expression for the log-likelihood. In our dataset, however, only four such measurements are present; they have a very weak influence on the results.

3. Last of all we sum the log-likelihoods over all the conditions $\mathcal{S}$ considered for the three different schemes to get the total log-likelihood:

$$\ln \mathcal{L}_{tot} = \sum_{\mathcal{S}} \ln \mathcal{L}(\mathcal{S}) \tag{16}$$

4. We maximize this global log-likelihood over the space of the nine parameters through the implementation of the *parallel tempering* algorithm, whose details are specified in appendix section Maximum likelihood fit procedure and *Appendix 1—figure 5* and *6*. We chose this algorithm because it ensures an effective search of the maximum even in a rugged parameter landscape.

5. Notice that the total log-likelihood $\ln \mathcal{L}_{tot}$ is sensitive to the number of measurements, which can vary considerably between different conditions. As such when performing the maximization the algorithm favors accuracy over the distributions with the higher number of measurements. Notice also that by evaluating the total likelihood in this manner we neglect the fact that multiple single cell measurements can come from the same stochastic realization of the process and can present some degree of correlation. To validate this inference procedure we generated 10 synthetic datasets using our stochastic model (see *Appendix 1—figure 11*), with the same number of measurements per scheme as in the experimental dataset. We then inferred, for each synthetic dataset, the values of the parameters, and compared them to their

groundtruth. On average all values of the parameters were correctly recovered, see appendix section Validation of inference procedure on artificially generated data and *Appendix 1— table 1*.

## Code and data availability

The code containing the implementation of our stochastic and deterministic model is made publicly available in the following repository: https://github.com/mmolari/affinity_maturation; (*Molari, 2020*; copy archived at https://github.com/elifesciences-publications/affinity_maturation). The repository also includes the experimental dataset, the code to run the inference procedure and the code to reproduce the figures of the main paper (*Figures 2–6*). Please refer to README.md file for further details.

# Acknowledgements

This work was supported by the CELLIGO project funded by the French government through BPI-France under the frame 'Programme d'Investissements d'Avenir' (PIA), the 'Institut Pierre-Gilles de Gennes' through the laboratoire d'excellence, 'Investissements d'avenir' programs ANR-10-IDEX-0001–02 PSL, ANR-10-EQPX-34, ANR-10-LABX-31 and ANR CE30-0021-01 (RBMPro). EK acknowledges generous funding from the 'The Branco Weiss Fellowship - Society in Science' and received funding from the European Research Council (ERC) under the European Union's Horizon 2020 research and innovation programme (Grant agreement No. 80336). We would like to further acknowledge the help of Dr. C Castrillon and Dr. P Bruhns for the work with the murine model system.

# Additional information

## Funding

| Funder | Grant reference number | Author |
|---|---|---|
| H2020 European Research Council | 80336 | Klaus Eyer |
| Agence Nationale de la Recherche | ANR-17-CE30-0021 RBMPro | Rémi Monasson |
| Agence Nationale de la Recherche | ANR-10-LABX-31 | Jean Baudry |
| Agence Nationale de la Recherche | ANR- 10-EQPX-34 | Jean Baudry |
| Agence Nationale de la Recherche | ANR-10-IDEX-0001-02 PSL | Jean Baudry Simona Cocco Rémi Monasson |

The funders had no role in study design, data collection and interpretation, or the decision to submit the work for publication.

## Author contributions

Marco Molari, Software, Formal analysis, Validation, Investigation, Visualization, Methodology, Writing - original draft; Klaus Eyer, Conceptualization, Data curation, Funding acquisition, Investigation, Methodology, Writing - original draft, Writing - review and editing; Jean Baudry, Conceptualization, Funding acquisition, Investigation, Methodology, Writing - original draft, Writing - review and editing; Simona Cocco, Rémi Monasson, Conceptualization, Formal analysis, Supervision, Funding acquisition, Investigation, Methodology, Writing - original draft, Project administration, Writing - review and editing

Author ORCIDs

Marco Molari (iD) https://orcid.org/0000-0001-8838-4093
Simona Cocco (iD) https://orcid.org/0000-0002-1852-7789
Rémi Monasson (iD) https://orcid.org/0000-0002-4459-0204

Ethics

Animal experimentation: Experiments using mice were validated by the CETEA ethics committee number 89 (Institut Pasteur, Paris, France) under #2013-0103, and by the French Ministry of Research under agreement #00513.02.

Decision letter and Author response

Decision letter https://doi.org/10.7554/eLife.55678.sa1
Author response https://doi.org/10.7554/eLife.55678.sa2

## Additional files

### Supplementary files

- Source data 1. Single-cell affinity measurement sheet.

- Transparent reporting form

### Data availability

All the data analysed in this work are reported in the supporting excel file attached to the submission. These data come from (1) new experiments reported in the present work, and (2) previously published experiments, see Eyer et al., 2017 (referenced in manuscript). The code containing the implementation of our stochastic and deterministic model is made publicly available in the following repository: https://github.com/mmolari/affinity_maturation (copy archived at https://github.com/elifesciences-publications/affinity_maturation). The repository also includes the experimental dataset, the code to run the inference procedure and the code to reproduce the figures of the main paper (Figures 2 to 6). Please refer to README.md file for further details.

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

## Appendix 1

## 1. Model definition and parameters choice

Mature GCs usually appear 5–7 days after Ag administration. During this time a population of up to hundreds different founder clones colonizes the GC and expands to a total size of a few thousand B-cells. The first mutations in the repertoire are observed around day 6 (*Jacob et al., 1993*; *McHeyzer-Williams et al., 1993*). Early GCs are highly polyclonal and contain 50 to 200 clones according to *Tas et al., 2016*. In agreement with these experimental findings at the time of Ag injection we pick a population of $N_{\mathrm{found}} = 100$ founder clones. The affinities of these clones are extracted independently from an initial gaussian distribution whose mean and variance are chosen via the maximum-likelihood procedure described in appendix sect. 4 ('Maximum likelihood fit procedure') and it matches the experimental distribution of germline responders (i.e. splenic IgG-SCs that are observed 1 day after boost of pure Ag, Appendix 1—figure 7 in scheme two and Ag dosage $D = 0$). During the time of GC formation the founder clones expand uniformly without mutating. We chose to start our simulation at $T_{\mathrm{GC}} = 6$ days after Ag injection. At this point, the GCs are almost fully formed (*De Silva and Klein, 2015*). The simulation starts with the GC at its maximal size, set to $N_i = N_{\max} = 2500$ clones. The maximal size is in agreement with (*Eisen, 2014*) which reports around 3000 cells per GC, or (*Tas et al., 2016*) in which GCs are said to contain up to a few thousands B-cells. However, we stress that GCs are heterogeneous in size (*Wittenbrink et al., 2011*).

From here the model proceeds in evolution rounds. Similarly to *Wang et al., 2015* we set the duration of a round to $T_{\mathrm{turn}} = 12\,\mathrm{h}$. This number is consistent with timing of cell migration (*Victora et al., 2010*; *Mesin et al., 2016*). We neglect the fact that high affinity cells are found to dwell longer in the GC dark zone (*Gitlin et al., 2014*) undergoing additional divisions. In addition to this the fact that the average cell-cycle time is 12 hr or longer (*Allen et al., 2007b*) indicates that 12 hr is probably a lower limit for the round duration.

As described in the main text each round consists in cell division with somatic hypermutation, selection for Ag binding, selection for T-cell help and differentiation. In our simulations before starting the first round we perform only once differentiation. This is done in order to recover the good average energy limit at low Ag concentrations. In fact when Ag dosage is small the population quickly goes extinct, while at the same time maturating very fast. Performing differentiation first provides a nucleus of low-affinity germline-like clones whose binding energy controls the average binding energy of the MC population, even if few high-affinity clones are added later. Notice that this does not change the asymptotic behavior of the model, since it would be equivalent to simply changing the order of operations in the round.

Proceeding with the standard turn order then the first operation performed is cell division and somatic hypermutation. During a round we consider cells to divide twice (*Mesin et al., 2016*). In GC dark zone cells up-regulate their expression of Activation-Induced Cytidine Deaminase. This enzyme increases the DNA mutation rate, inducing mutations in the region coding for the BCR and possibly changing the affinity for the Ag. Mutation rate has been estimated to an average of $10^{-3}$ mutations per base pair per division (*McKean et al., 1984*; *Kleinstein et al., 2003*). Similarly to *Wang et al., 2015* in which the total binding energy consisted in the sum of contributions from 46 different residues, we consider $N_{\mathrm{res}} = 50$ residues to contribute to the binding energy. The probability that upon division at least a mutation occurs in any of the 150 bp coding for these residues can be estimated as $p_{\mathrm{mut}} = 1 - (1 - 10^{-3})^{150} \sim 0.14$. As done in *Wang et al., 2015*; *Zhang and Shakhnovich, 2010*; *Wang, 2017* at every division and for each daughter cell independently we consider a $p_{\mathrm{sil}} = .5$ probability of developing a silent mutation, in which case the binding energy of the daughter cell remains unchanged, a probability $p_{\mathrm{let}} = .3$ of undergoing a lethal mutation, in which case the cell is removed, and finally a probability $p_{\mathrm{aa}} = .2$ of developing an affinity-affecting mutation. These change the binding energy of the daughter cell by adding a variation $\epsilon \rightarrow \epsilon + \Delta\epsilon$. As done in *Wang, 2017* the variation follows a lognormal distribution $K_{\mathrm{aa}}(\Delta\epsilon)$ (*Appendix 1—figure 1A*)

$$\text{Lognorm}[\mu, \sigma](x) = \begin{cases} \frac{1}{x\sigma\sqrt{2\pi}}\exp\left\{-\frac{(\ln x - \mu)^2}{2\sigma^2}\right\} & x > 0 \\ 0 & x \leq 0 \end{cases} \tag{17}$$

$$K_{\text{aa}}(\Delta\epsilon) = \text{Lognorm}[\mu = 1.9, \sigma = 0.5](\Delta\epsilon + 3) \tag{18}$$

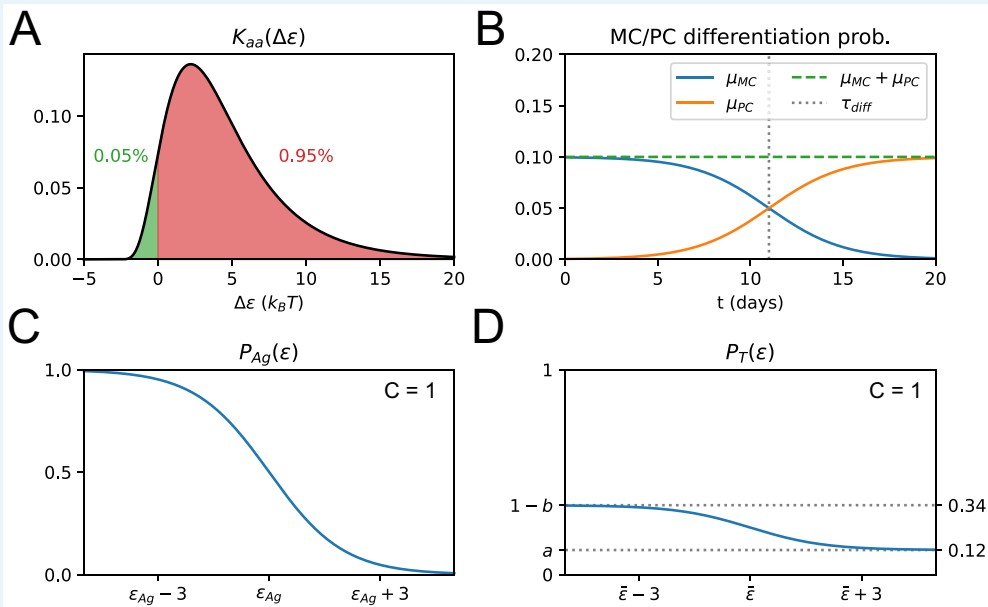

**Appendix 1—figure 1.** Effect of affinity-affecting mutations, differentiation time-switch and survival probability. (**A**) Plot of the affinity-affecting kernel $K_{aa}$. Only around 5% of the affinity-affecting mutations increase affinity (green part of the curve for $\Delta\epsilon < 0$). (**B**) Probability of MC $\mu_{MC}$ and PC $\mu_{PC}$ differentiation as a function of time from Ag injection ($t$ in days). (**C**) Probability $P_{\text{Ag}}(\epsilon)$ that a B-cell survives Ag-binding selection process as a function of its binding energy $\epsilon$, see *Equation 19*. On the x-axis we mark the threshold energy $\epsilon_{\text{Ag}}$. (D) Probability $P_{\text{T}}(\epsilon|\bar{\epsilon})$ that a B-cell survives the T-cell selection process as a function of its binding energy $\epsilon$, see *Equation 20*. On the x-axis we mark the threshold binding energy $\bar{\epsilon}$, which depends on the binding energy distribution of the rest of the B-cell population, see *Equation 20*.

The parameters of the distribution are chosen so that only 5% of the mutations confer an increase in affinity, while the vast majority causes an affinity decrease. As a result of this process after the two mutations the population size increases almost 4-fold in size (two duplications but some cells are eliminated due to lethal mutations) and the average affinity decreases slightly due to the mainly negative effect of mutations (cf *Figure 1* main text, histograms 1 to 2).

After duplication we implement selection. In order to avoid apoptosis cells must bind and internalize a sufficient amount of Ag. The amount of Ag internalized depends both on the affinity of the BCR and on the availability of Ag on the surface of the FDC (*Nowosad et al., 2016*; *Knežević et al., 2018*; *Batista and Neuberger, 2000*). We model this process by expressing the probability of survival of a cell with BCR having binding energy $\epsilon$ as described in the main text:

$$P_{\text{Ag}}(\epsilon) = \frac{Ce^{-\epsilon}}{Ce^{-\epsilon} + e^{-\epsilon_{\text{Ag}}}} \tag{19}$$

A sketch of this function is shown in *Appendix 1—figure 1C*. The value of the threshold binding energy has been obtained via maximum likelihood fit of the data, which yields for example $\epsilon_{\text{Ag}} = -13.59$ for variant A. This selection is not in present variant C of the model.

At the second step of selection, the one leading maturation in our model, B-cells compete to receive a survival signal from T-cells. T-cells in GCs are motile and continuously scan the surface of B-cells, sensing for density of pMHC-II complexes (**Shulman et al., 2014**). Cells with the highest pMHC-II density receive survival signal preferentially (**Depoil et al., 2005**; **Victora and Nussenzweig, 2012**). We again express the probability of survival through

$$P_{\rm T}(\epsilon|\bar\epsilon) = a + (1-a-b)\frac{Ce^{-\epsilon}}{Ce^{-\epsilon}+e^{-\bar\epsilon}}, \quad \text{with} \quad e^{-\bar\epsilon} = \langle e^{-\epsilon}\rangle_{GC}, \tag{20}$$

The parameters $a$ and $b$ in **Equation 20** represent, respectively, the probability of survival at very high energy and the deficit in probability of survival at very low binding energy. Their effect is better discussed in appendix section (Permissive and stochastic selection: effect of $a$, $b$ parameters). The formula interpolates smoothly between these two values, as depicted in **Appendix 1—figure 1D**. The threshold binding energy $\bar\epsilon$ depends on the population's binding energy distribution, introducing competition between the cells; the symbol $\langle\cdot\rangle_{GC}$ indicates the average of the quantity over the current GC population.

Cells that are able to survive selection can either re-enter the dark zone and start a new round of evolution or differentiate into Ab-producing PCs or quiescent MCs that can be reactivated upon future Ag injection. There is evidence that MC/PC output undergoes a temporal switch: MCs are preferentially produced early in the response (**Weisel et al., 2016**). Moreover there seems to be an affinity bias in differentiation (**Shinnakasu and Kurosaki, 2017**). Even though experiments show that affinity plays a role in deciding fate (**Shinnakasu et al., 2016**) simply by implementing a time-switch in the MC/PC differentiation probability (respectively $\mu_{\rm MC}$, $\mu_{\rm PC}$, **Appendix 1—figure 1B**) we effectively recover both of these observations. The parameters of these functions ($\tau_{\rm diff} = 11d$, $\Delta\tau_{\rm diff} = 2d$) are chosen so as to be compatible with **Weisel et al., 2016**:

$$\mu_{\rm MC}(t) = p_{\rm diff}\frac{1}{1+\exp\{\frac{t-\tau_{\rm diff}}{\Delta\tau_{\rm diff}}\}} \tag{21}$$

$$\mu_{\rm PC}(t) = p_{\rm diff}\frac{1}{1+\exp\{-\frac{t-\tau_{\rm diff}}{\Delta\tau_{\rm diff}}\}} \tag{22}$$

Notice that the sum of the two is constant $\mu_{\rm MC}(t)+\mu_{\rm PC}(t) = p_{\rm diff} = 10\%$, compatible with seminal studies (**Oprea and Perelson, 1997**) that estimated that around 90% of the cells recirculate in the dark zone. In the model we consider for simplicity a complete switch, meaning that for $t \gg \tau_{\rm diff}$ the probability of generating MCs decreases asymptotically to zero. In appendix section (Permissive and stochastic selection: effect of $a$, $b$ parameters), we discuss the more realistic case of a partial switch, in which there is a residual probability of MC production even for $t \gg \tau_{\rm diff}$.

If new Ag is administered, we consider a new GCR to start. The new GC is colonized partly by new B-cells coming from the naive pool and partly by reactivated MCs (**Inoue et al., 2018**). We allow only MCs that have already been generated at time of the second injection to colonize the new GC. This is done by picking a set of $N_i = 2500$ cells from the naive pool, with binding energies extracted from the same initial Gaussian distribution, and adding to them all the MCs generated up to the time of second injection. The founder clones of the new GC will consist of $N_{found} = 100$ cells randomly extracted from this cumulative population. Notice that the probability of extracting a MC from the cumulative population is an increasing function of the number $N_{\rm MC}$ of MCs extracted at time of injection: $p = N_{\rm MC}/(N_i + N_{\rm MC})$. In appendix sect. 12 we discuss instead the case in which the probability of extracting a seeder clone from the memory pool is set to a constant $p = 0.3$.

The concentration of Ag evolves as explained in the main text according to the differential **Equations 5 and 6** (main text). These equations account for Ag release, decay and consumption. The release rate was evaluated considering a half-life of 17 hr for Ag in CFA (**MacLean et al., 2001**), which gives a value for the release rate of $k^+ = \ln 2/\tau_{1/2} \sim 0.98/{\rm d}$. Ag on FDCs can be maintained for a long time, up to a year (**Heesters et al., 2014**), through a

mechanism of endocytosis and recycling of immune complexes (*Heesters et al., 2013*). To reproduce this long clearance time, we take Ag lifetime to be 8.1 weeks, as measured in popliteal lymph nodes of mice (*Tew and Mandel, 1979*). This results in a Ag decay rate of $k_{\varnothing}^- = \ln 2/\tau_{1/2} \sim 0.012\,/\mathrm{d}$. The case of a faster Ag-decay is discussed in appendix sect. 6 ('Possible model variations'). Finally, the consumption rate per B-cell $k_B^- = 2.07 \times 10^{-5}\,/\mathrm{d}$ (variant C, see main text) is obtained via the maximum likelihood fit procedure described in appendix sect. 4 ('Maximum likelihood fit procedure'). This quantity controls both the GC lifetime and the extent of AM at the end of evolution. For the range of Ag dosages considered simulated GCs have an effective lifetime that vary between 1 or 2 weeks and 3 months (*Appendix 1—figure 2E*), compatible with lifetimes of real GCs (*Victora and Mouquet, 2018*). *Equations 5 and 6* (main text) are continuous in time. To include them in our discrete timestep model, we perform an update of the values of the reservoir and available concentrations $C_{\mathrm{av}}(t)$, $C_{\mathrm{res}}(t)$ at each round $t = 0, 1, \ldots$ after selection for T-cell help and before differentiation. The Ag removal rate is given by the cumulative effect of decay and consumption: $k_t^- = k_{\varnothing}^- + N_t^B k_B^-$, and changes at each evolution round due to its dependence on the number of B-cells $N_t^B$ at this stage of the round. The values of the concentrations at the next round $t+1$ are obtained by evolving the corresponding quantities at round $t$ for a time $T = 12\,\mathrm{h}$ equivalent to the duration of the round:

$$C_{\mathrm{res}}(t+1) = C_{\mathrm{res}}(t)\, e^{-k^+ T} \tag{23}$$

$$C_{\mathrm{av}}(t+1) = C_{\mathrm{av}}(t)\, e^{-k_t^- T} + C_{\mathrm{res}}(t)\, \frac{k^+}{k^+ - k_t^-} \left( e^{-k_t^- T} - e^{-k^+ T} \right) \tag{24}$$

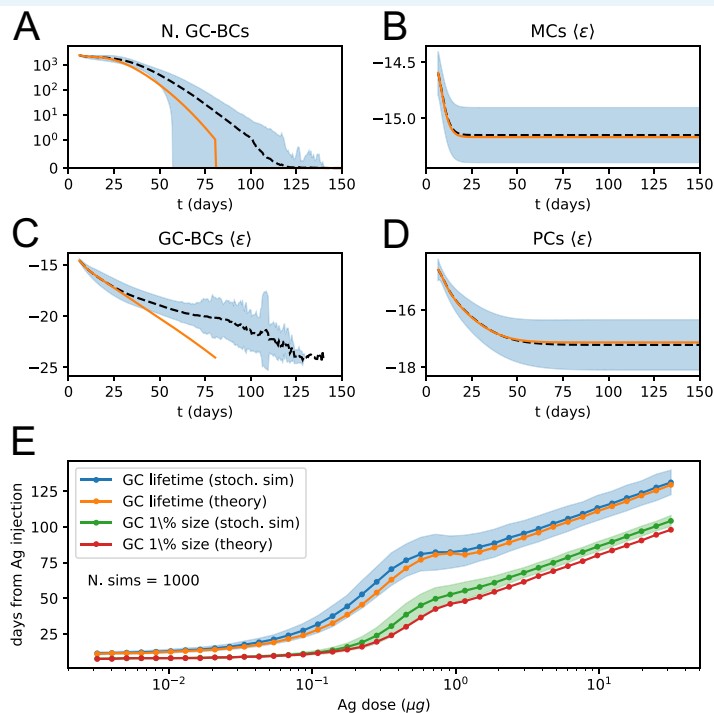

**Appendix 1—figure 2.** Deterministic and stochastic evoliution comparison. (**A to D**) Comparison between average evolution of the system and prediction of our deterministic model. The average is performed over 1000 independent GCR simulations at injected dosage of Ag $D = 1\mu g$. The black dashed line corresponds to the average stochastic trajectory, the shaded blue area covers one standard deviation from the trajectory mean, and the orange lines correspond to the prediction of the deterministic model. Number of GC B-cells (**A**) and average energy of GC B-cells (**C**), MCs (**B**) and PCs (**D**) as a function of time from Ag injection. (**E**) Average lifetime and effective lifetime of GCs as a function of administered Ag dosage.

The latter corresponds to the time at which the GC decays to 1% of its initial size. We compare the average over 1000 stochastic simulations (blue and green plots, shaded area corresponds to one standard deviation from the mean) to the theoretical prediction (orange and red lines). Due to finite-size effects, the theory in general slightly underestimates the lifetime of the GC.

At times smaller than the GC formation time $T_{\mathrm{GC}} = 6\,\mathrm{d}$ we do not account for GC evolution but we account for the evolution of concentration. This is done as in the previous equations but in this case the total consumption rate is evaluated considering an exponentially increasing number of cells, that starting from one at injection exponentially grows to $N_{\mathrm{max}}$ at the time of GC formation. In particular concentration update at the end of round $t = 0, 1, \ldots, 11$ is done considering the following number of B-cells consuming Ag:

$$N_t^B = N_{\mathrm{max}}{}^{t \times T_{\mathrm{turn}}/T_{\mathrm{GC}}} \quad \mathrm{for} \quad t < 12 = T_{\mathrm{GC}}/T_{\mathrm{turn}} \tag{25}$$

In our simulations, GC evolution stops either naturally when Ag depletion leads to population extinction, or when the total simulation time is elapsed and cells are harvested, in which case the simulation is stopped irrespective of the population size and only cells produced up to that point are considered. The total simulation time depends on the immunization scheme considered and is set to match the time elapsed between injection and experimental measurement.

## 2. Stochastic model analysis

In order to have some insight on the way our stochastic model evolves and to gauge the magnitude of stochastic effects in *Appendix 1—figure 2A to D* we report the average evolution of 1000 independent GC Reaction (GCR) simulations, performed at an injected Ag dosage of $D = 1\mu g$. The average trajectory is reported as a black dashed line and shaded area covers one standard deviation around the mean. Increase of noise in the average binding energy of GC B-cells as a function of time is due to the fact that at each time the average is performed only over the surviving GCs, thus as progressively GCRs end the average becomes more noisy (and also becomes biased to represent only surviving trajectories). We compare the evolution of the stochastic model with the theoretical prediction (orangle lines). As observed in the main text the theoretical prediction performs well at high population size, and loses accuracy for small population sizes (*Appendix 1—figure 2A, C*). However, since most of the MC and PC population is generated at time of high population expansion the average energy of these two populations is always well estimated (*Appendix 1—figure 2B, D*). The inaccuracy at small population sizes comes mainly from an overestimation of the selection pressure in our theoretical solution. In fact, the threshold binding energy for T-cell selection $\bar{\epsilon}$ (cf appendix *Equation 20*) is sensitive to the high-affinity tail of the population affinity distribution. As the population size diminishes this tail gets progressively less populated and the value of $\bar{\epsilon}$ deviates from its theoretical prediction, evaluated under the infinite-size limit. This slight decrease of selection pressure at the end of the GC lifetime increases slightly the survival time and generates a slight slow-down in maturation. Moreover, in order to estimate the lifetime of GCs in our model we evaluate average and standard deviation of lifetimes of 1000 independent GC simulations for different values of injected Ag dosage. Since in the simulations GCs can also have a more or less long period of small population size prior to extinction (*Appendix 1—figure 2A*) we also evaluate an 'effective' lifetime, considering the GC effectively extinguished when its size reaches 1% of its original size. These lifetimes (*Appendix 1—figure 2E*) depend on the amount of Ag administered and can vary between few weeks to few months. The same overestimation of the selection pressure at small population sizes leads the theory to slightly underestimate the lifetime of stochastic simulations.

Recent experiments estimated the number of different clones in early GC to be between 50 and 200 (*Tas et al., 2016*). In our stochastic model, we consider the population of founder clones to be composed of 100 cells. The limited number of founders controls the diversity of the initial population and increases the stochasticity in evolution. However, it does not strongly

influence the average outcome. To verify this we compare 1000 stochastic simulations of the standard model (single Ag injection of $D = 1\,\mu g$ of Ag, model scenario C) with a modified version in which the number of founder clones was set equal to the number of cells in the initial population (2500 cells). Results are reported in *Appendix 1—figure 3A to D*. We observe that limiting the initial population diversity increases stochasticity in evolution, but does not impact much the average evolution trajectory and outcome. This is especially evident when observing MC/PC population evolution (panels B and D). The final average binding energies of these populations are very similar, but the standard deviation around the mean is halved in for the initial population with more founders.

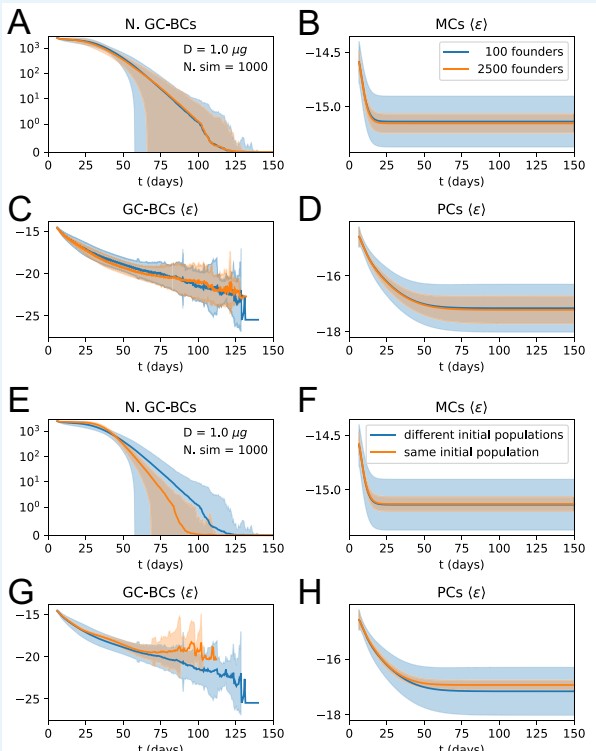

**Appendix 1—figure 3.** Influence of number and affinity of founder clones on evolution stochasticity. (**A to D**) Effect of increasing the number of founder clones in the population. We compare 1000 stochastic simulations of the standard version of the model (blue) with a modified version in which every cell in the initial population (2500 cells total) originates from a different founder clone, and has therefore a different affinity (orange). Solid lines represent average trajectories and the shaded area covers one standard deviation from the mean. The plots represent the number of GC B-cell (**A**), their average binding energy (**C**), and the average binding energy of MC (**B**) and PC (**D**) population as a function of time from Ag injection. (**E** to **H**) Stochastic contribution of the initial founder clones population. We compare 1000 stochastic simulations of the standard version of the model (blue) with a modified version in which GCs are initialzied with the same 100 founder clones (orange). We observe that the initial choice of founder clones plays an important role in evolution and explains most of the variation observed in the outcome. Figures E to H display the same quantities as A to D.

This observation raises the question of how much the outcome of evolution is controlled by the particular initial choice of the founder clones. In *Appendix 1—figure 3E to H* we quantify this by comparing 1000 stochastic GC evolutions of the standard model (injected Ag dosage $D = 1\,\mu g$, model scenario C), in which the founder population was re-extracted every time, with a modified version in which the founder population was kept the same amongst all stochastic trials. In the latter case, we observe a considerable reduction in stochasticity, indicating that the outcome depends strongly on the initial founder clones choice. This is also

in line with the observations made in section 'Maturation induces progressive loss of clonality' and *Figure 3D* (main text), where we show that the presence of a high-affinity founder clones correlates with a stronger homogenizing selection.

## 3. Theoretical solution and eigenvalue equation

As described in the main text the theoretical solution of the model is obtained by performing the limit of infinite size $N_B \to \infty$, upon which the evolution of the system becomes deterministic. All the stochastic processes can be implemented via an operator, whose explicit expression is provided in Materials and methods, and which acts on the distribution $\rho(\epsilon)$. This function is the product between the binding energy distribution of the population and the population size. In this formalism, the total evolution operator is a combination of the duplication and mutation, selection, normalization and differentiation operators: $\mathbf{E} = \mathbf{D} \, \mathbf{N} \, \mathbf{S}_\mathrm{T} \, \mathbf{S}_\mathrm{Ag} \, \mathbf{R}$, where the operator $\mathbf{R} = \mathbf{M} \, \mathbf{A} \, \mathbf{M} \, \mathbf{A}$ encodes for two rounds of duplication with mutation. The mutation operator $\mathbf{M}$ in particular consists in a convolution of the distribution $\rho$ with a mutation kernel $K_\mathrm{eff}(\Delta\epsilon)$ lethal, silent and affinity-affecting nature of the mutations. This kernel is defined as follows:

$$K_\mathrm{eff}(\Delta\epsilon) = p_\mathrm{mut} \, p_\mathrm{aa} \, K_\mathrm{aa}(\Delta\epsilon) + (1 - p_\mathrm{mut} + p_\mathrm{mut} \, p_\mathrm{s}) \, \delta(\Delta\epsilon) \tag{26}$$

The first term corresponds to affinity-affecting mutations, whose probability is the product between the probability of developing a mutation and the probability for this mutation to be affinity-affecting $p_\mathrm{mut} \, p_\mathrm{aa}$. The expression for the affinity-affecting mutation probability $K_\mathrm{aa}(\Delta\epsilon)$ is the one in *Equation 18*. The second term encodes silent mutations, occurring with probability $p_\mathrm{mut} \, p_\mathrm{s}$, and also absence of mutations, with probability $1 - p_\mathrm{mut}$. Lethal mutations occurs with probability $p_\mathrm{mut} \, p_\mathrm{l}$ but since their effect is the removal of cells in the population this contribution is multiplied by zero and is not present in the kernel expression. The contribution of lethal mutation makes so that the normalization of this kernel is not unitary: $\int d\Delta\epsilon \, K_\mathrm{eff}(\Delta\epsilon) = 1 - p_\mathrm{mut} \, p_\mathrm{l}$. In the main text, we show how under constant Ag concentration the affinity distribution of the population behaves as a traveling wave whose velocity and growth rate can be found via an eigenvalue equation. This eigenvalue equation reads $\Sigma(-u)\mathbf{E}\rho = e^\phi \rho$, where $\Sigma(\Delta)$ is the translation operator that shifts the distribution of a value $\epsilon \to \epsilon + \Delta$ and we consider the restricted evolution operator $\mathbf{E} = \mathbf{D} \, \mathbf{S}_\mathrm{T} \, \mathbf{R}$.

To verify the correctness of our eigenvalue equation theoretical prediction, we solve the eigenvalue problem at a given Ag concentration $C = 10$, graphically illustrating the procedure, and show that the theoretical prediction matches the asymptotic behavior of the system. As a first step we set the value of $\bar\epsilon = 0$ in the T-cell help selection operator. This choice constitutes simply a gauge-fixing that removes the translational invariance of our solution, and it also linearizes the evolution operator. Since we do not a priori know the value of the shift $\Delta$ we solve the eigenvalue equation $\Sigma(-\Delta)\mathbf{E}\rho = e^\phi \rho$ for different values of the shift. In *Appendix 1— figure 4* we plot the maximum eigenvalue eigenvectors (distributions in A, color encodes the value of $\Delta$) and their corresponding log-eigenvalues (C, representing the growth rate) as a function of the shift $\Delta$. In order for our solution to be consistent it must satisfy the condition $\bar\epsilon = 0$. Therefore we evaluate $\bar\epsilon$ after performing mutations for all the solutions at varying values of the shift (*Appendix 1—figure 4B*) and we pick the one ($\Delta^*$) for which this condition is satisfied as the eigenproblem solution. Upon repeated application of the restricted evolution operator we expect the population to asymptotically grow and shift at the values of $\phi^*$ and $u = \Delta^*$ corresponding to this solution. In *Appendix 1—figure 4D*, we verify this by plotting the evolution of the normalized affinity distribution of the population, re-shifted on its mean, upon repeated application of the evolution operator, and compare it with the eigenproblem solution (black dashed lines). Moreover, in E and F we compare the instantaneous growth rate and energy shift with the eigenproblem solution predictions. Color in the three plots encodes to the evolution round. We observe that in all three cases the asymptotic prediction is matched.

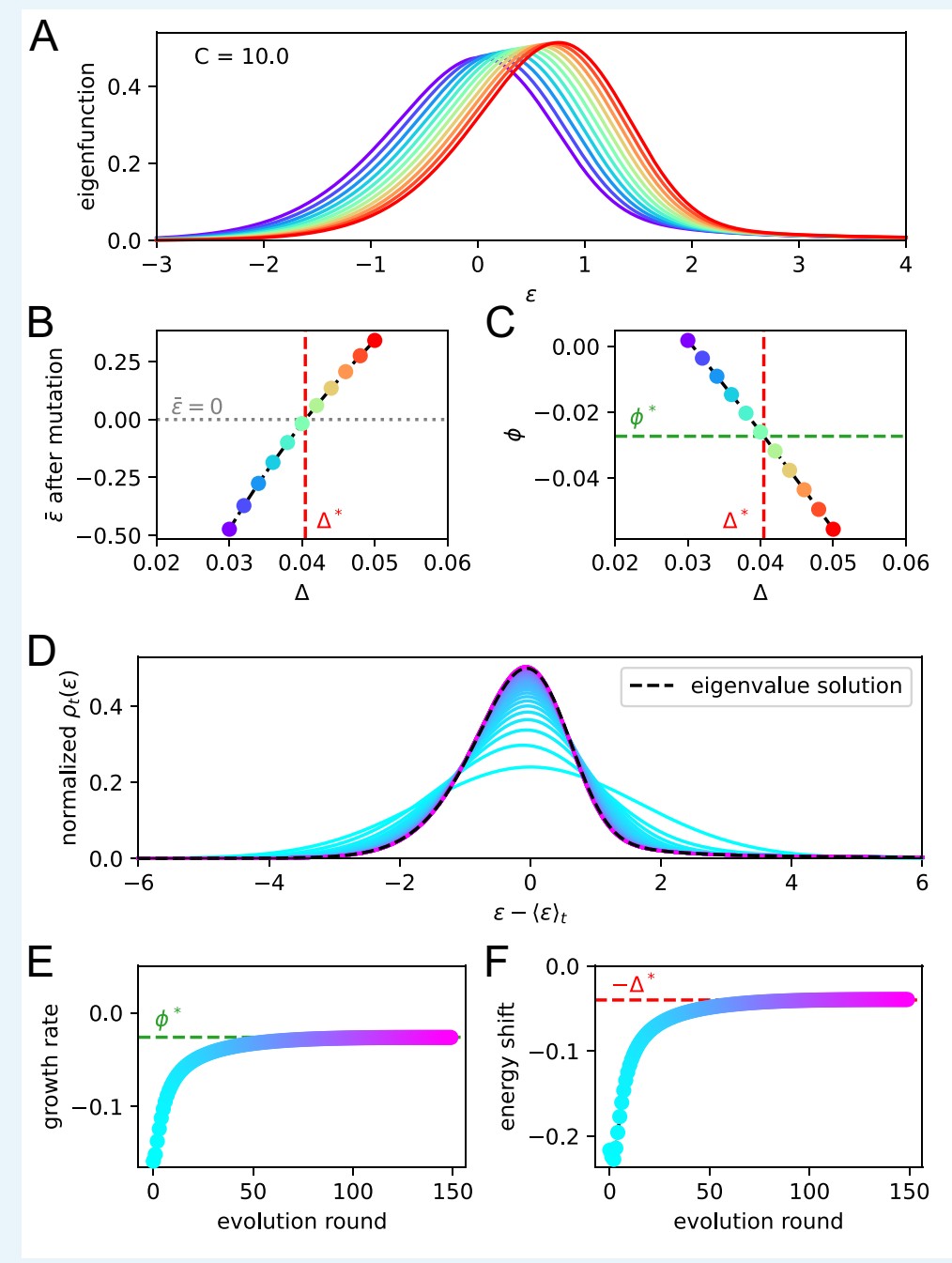

**Appendix 1—figure 4.** Asymptotic evolution matches the solution of the eigenvalue equation. We check that upon repetition of the evolution operator the system converges at the eigenvalue equation solution. For a given constant Ag concentration ($C = 10$ in our case) we solve the eigenvalue equation $e^{\phi}\rho = \Sigma(-\Delta)\mathbf{E}\rho$ for various values of the shift $\Delta$. In A we report the maximum eigenvalue eigenfunctions. By virtue of the Perron-Frobenius theorem these consist of only positive values. Color represent the value of the shift $\Delta$ for the corresponding solution. In B and C, we plot the value of $\bar{\epsilon}$ after mutation and of the growth rate $\phi$ for every solution. The consistency condition requires us to pick the eigenfunction for whom the value of $\bar{\epsilon}$ after mutation is zero. This corresponds to the value $\Delta^*$ represented in vertical red dashed line and the value of the growth rate $\phi^*$ in horizontal green dashed line. In panels D, E, F, we consider repeated application of the evolution operator to the binding energy distribution at constant Ag concentration $C = 10$. Color encodes the number of applications of this operator.

In D and E, we report respectively the growth rate and shift of the mean per evolution turn, and in F the full distribution of binding energies, normalized to the population size. All the quantities converge to their theoretical expectation given by the chosen solution of the eigenvalue equation, reported as green and red dashed lines.

To numerically solve the eigenvalue equation and obtain the results displayed in *Appendix 1—figure 4*, operators were implemented as square matrices by discretizing the interval of energy $[10, -20]\, k_B T$ with a discretization step of $0.002\, k_B T$. The maximum eigenvalue and corresponding eigenvector were obtained by repeated application of the restricted evolution and shift operators on an initially normal distribution with zero mean and unit variance. The application was repeated until the L1 distance between the normalized distributions before and after application of the operator was less than $10^{-5}$. This numerical procedure however is computationally expensive, since solving the eigenproblem for a particular value of the Ag concentration $C$ requires the creation and repeated application of large matrices for many values of the shift $\Delta$. A less demanding numerical method to obtain the value of $\mu$ and $\phi$ consists in simply simulating the evolution of the population under constant Ag concentration $C$, and without carrying capacity constraint and Ag-binding selection, until convergence to the asymptotic travelling-wave state is reached with sufficient precision. This second technique was used to evaluate the values of $u$ and $\phi$ displayed in *Figure 4B*. In this case, the simulation domain was set to the interval $[-100, 50]\, k_B T$, with a discretization step of $0.01\, k_B T$. Evolution of the distribution of binding energies was repeated until the following three conditions were met at the same time. First, the L1 distance between the normalized distributions before and after evolution, once they were re-centered around their mean, was less than $5 \times 10^{-5}$. Second, the relative change of the growth rate and wave speed between two rounds of evolution was less than $5 \times 10^{-5}$. Finally, as a safety check we also require that at the moment of convergence the mean of the distribution is more than five standard deviations away from the boundaries of the simulation domain.

## 4. Maximum likelihood fit procedure

Nine model parameters $\mathbf{p} = (\mu_{\text{naive}}, \sigma_{\text{naive}}, \epsilon_{\text{Ag}}, k_B^-, \alpha, a, b, g_{\text{recall}}, g_{\text{imm}})$ were inferred via the maximum likelihood procedure described in Materials and methods. This procedure makes use of the full experimental affinity distribution. Here, we give a more detailed description of the procedure and add considerations on the variation range of the parameters.

The stochastic maximization procedure is based on an implementation of the *Parallel Tempering* technique. This technique, used in the context of molecular dynamics simulations (*Sugita and Okamoto, 1999*), consists in simulating different copies of a system at different temperatures, and then allowing the copies to exchange their states with adequate probabilities so that low-energy states are correctly sampled at low temperatures. This is particularly advantageous when the energy landscape is rugged, and low-temperature simulations tend to get stuck in local energy minima, while high-temperature simulation explore the whole space without being able to locate the minima precisely. Allowing for state-exchange between different temperature simulations makes so that high-temperature simulations can help low-temperature ones exit local minima in the energy space and converge to the global optimum, *Appendix 1—figure 5A*.

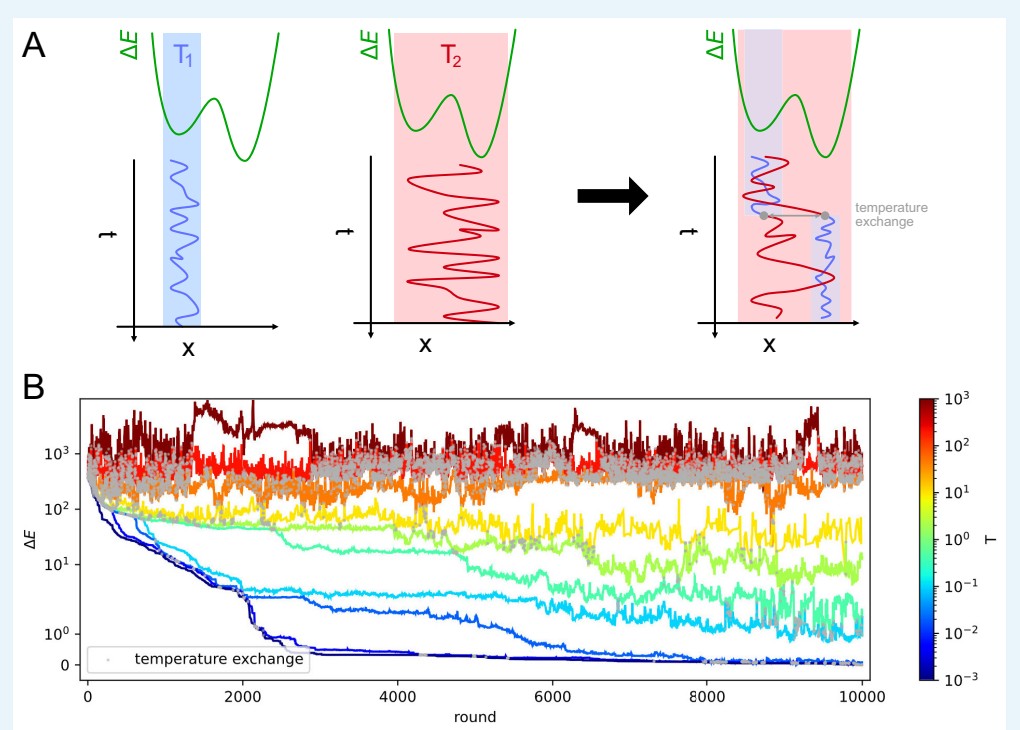

**Appendix 1—figure 5.** Parallel tempering helps exiting local minima in stochastic likelihood maximization. **A**: intuitive representation of the advantage of Parallel Tempering. When a Monte Carlo simulation is run at low temperature ($T_1$) the system reaches a low-energy state but can get stuck in local energy minima. Conversely at high temperature ($T_2$) the system is free to explore a larger portion of the space, but is unable to localize the energy minimum. By allowing the states to exchange their temperature when favorable, the high temperature simulation can help the low temperature simulation exit from a local minimum trap and converge to the global energy minimum. **B**: Simulation trajectories in energy space as a function of the simulation round for inference on variant C of the model. The energy difference is defined as $\Delta E = -(\log \mathcal{L} - \log \mathcal{L}_{\max})$, where $\mathcal{L}_{\max}$ is the maximum likelihood recovered by the inference algorithm. Colors encode different temperatures according to the colorbar on the right. Grey dots mark points in which trajectories exchange temperatures. To display both large variations and values equal to zero the energy scale is logarithmic for values of $\Delta E > 1$ and linear for energies $0 \leq \Delta E \leq 1$.

Here, we consider the space of all possible values of the parameters as our configuration space, and use the log-likelihood (see Materials and methods for definition) as a proxy for minus the energy. Our algorithm can be summarized as follows. A number $N = 10$ of copies of the parameter set is initialized, and at each copy is assigned a simulation temperature $T$ logarithmically evenly spaced between $10^3$ and $10^{-3}$. The maximization procedure consists in 10,000 rounds of iterative parameter changes and temperature exchanges. For each proposed parameter change, the likelihood difference $\Delta \log \mathcal{L}$ between the new and the original set of parameters is evaluated, and the change is accepted with probability $\min\{\exp\{\Delta \log \mathcal{L}/T\}, 1\}$, where $T$ is the temperature associated to the parameter set. In the exchange phase the difference in log-likelihood $\Delta \log \mathcal{L}$ and in inverse temperature $\Delta \beta$ is evaluated for any two parameters sets with consecutive temperatures. An exchange of the two parameters sets is then performed with probability $\min\{\exp\{-\Delta \beta \, \Delta \log \mathcal{L}\}, 1\}$. At the end of the last round the value of the parameters $\mathbf{p}_{\text{best}}$ that maximized the likelihood is returned. In **Appendix 1—figure 5B** we report the trajectories in energy space of all the parameter sets for the inference performed using variant C of selection (see main text), in which eight parameters are inferred (all but the threshold binding energy $\epsilon_{\text{Ag}}$, which is removed in this variant). The energy for each parameter set is evaluated from the difference in log-likelihood with the best value,

$\Delta E = \log \mathcal{L}(\mathbf{p}_{\text{best}}) - \log \mathcal{L}(\mathbf{p})$. Notice how trajectories at high temperature explore the space by being able to visit low-likelihood (high-energy) zones of the parameter space. Conversely, low-temperature trajectories gradually converge to the value of the parameters that maximizes the likelihood.

The likelihood-maximization procedure is algorithmically described by the following pseudocode:

---

**Algorithm:** Stochastic likelihood maximization procedure

---

Given the initial parameters choice $\mathbf{p}_{\text{init}}$;
Initialize 10 copies of the system with parameter value $\mathbf{p}_i^0 = \mathbf{p}_{\text{init}}$ for $i = 1 \dots 10$;
Set the simulation temperature of each copy of the system at $T_i = 10^{(11-2i)/3}$, so that $T_1 = 10^3$ and $T_{10} = 10^{-3}$;
for $t = 1$ to 10'000 do:
 for $i = 1$ to 10 do:
 Generate a new randomly mutated parameter set $\mathbf{p}_i' = \mathbf{p}_i^{t-1} + \Delta \mathbf{p}_i$
 according to the rule specified in the text;
 Evaluate the log-likelihood difference induced by
 the new set of parameters $\Delta \log \mathcal{L} = \log \mathcal{L}(\mathbf{p}_i') - \log \mathcal{L}(\mathbf{p}_i^{t-1})$ ;
 With probability $\min\{\exp\{\Delta \log \mathcal{L}/T_i\}, 1\}$ accept $\mathbf{p}_i^t \leftarrow \mathbf{p}_i'$ or else keep $\mathbf{p}_i^t \leftarrow \mathbf{p}_i^{t-1}$ ; for
 for $i$ to 9 do
 Evaluate the likelihood difference $\Delta \log \mathcal{L} = \log \mathcal{L}(\mathbf{p}_{i+1}^t) - \log \mathcal{L}(\mathbf{p}_i^t)$ between
 two adjacent copies of the model;
 Evaluate the inverse temperature difference $\Delta \beta = 1/T_{i+1} - 1/T_i$;
 With probability $\min\{\exp\{-\Delta \beta \, \Delta \log \mathcal{L}\}, 1\}$ perform the state exchange $\mathbf{p}_i^t \leftrightarrow \mathbf{p}_{i+1}^t$;
Find $(i^*, t^*) = \arg\max_{(i,t)}\{\log \mathcal{L}(\mathbf{p}_i^t)\}$;
return $\mathbf{p}_{\text{best}} = \mathbf{p}_{i^*}^{t^*}$

---

Where the initial parameters choice was set to

$\mathbf{p}_{\text{init}} = (\mu_{\text{naive}} = -14.6, \sigma_{\text{naive}} = 1.6, \epsilon_{\text{Ag}} = -13.6, k_B^- = 2 \times 10^{-5} \,/\text{d}, \alpha = 0.025 \,\mu\text{g}, a = b = 0.2,$
$g_{\text{recall}} = g_{\text{imm}} = 0.5)$. Single parameter changes are generated as a function of noise level $\eta$:

- For $\mu_{\text{naive}}, \epsilon_{\text{Ag}}$ variation is performed by adding a random number extracted with uniform probability in the interval $[-10\,\eta, +10\,\eta]$.
- For positive parameters $\sigma_{\text{naive}}, \alpha, k_B^-$ the variation is performed multiplicatively by introducing a percentage change of the parameter uniformly extracted in the interval $[-\eta, \eta]$.
- For the fractions $g_{\text{recall}}, g_{\text{imm}}, a$ and $b$ variations are performed by adding a random number uniformly extracted in the interval $[-\eta, +\eta]$. After the addition some constraints are enforced: the resulting number is constrained in the interval $[0, 1]$, and as a consequence of the definition of the survival probability (**Equations 19, 20**) also $a + b \leq 1$ is imposed.

We set the noise level to depend on the temperature of the system considered, so that higher-temperature simulations also have a higher level of noise, allowing them a faster exploration of the parameter space, while low temperature simulation will perform only a fine-tuning of the parameters. This allows for a more precise convergence. In particular we set $\eta$ to be logarithmically evenly spaced between $10^{-2}$ and $10^{-1}$. Furthermore we propose changes for all the parameters for the four higher-temperature simulations, while changes affect only one randomly chosen parameter for the four lower-temperature simulations.

In **Appendix 1—figure 6**, we display the evolution of the parameter set that reaches the highest log-likelihood during the maximization procedure. In **Appendix 1—figure 6A** we report the log-likelihood and the temperature as a function of the round of maximization. Notice how high-temperature states correlate with big fluctuations in the log-likelyhood and in the values of parameters. The peak likelihood value is reached at round 9970, when the trajectory was at the lowest temperature. In **Appendix 1—figure 6B** we report the evolution of the nine parameters during the maximization rounds. Orange shaded area cover one standard deviation of the posterior distribution around the maximum-likelihood estimate (MLE) of the parameters. This was obtained by approximating the posterior with a Gaussian distribution around the ML value, with a quadratic fit of the log-likelihood variation $\Delta \log \mathcal{L}$ generated by small variations ($\pm 5\%$) of single parameters around the MLE (**Appendix 1— figure 6C**).

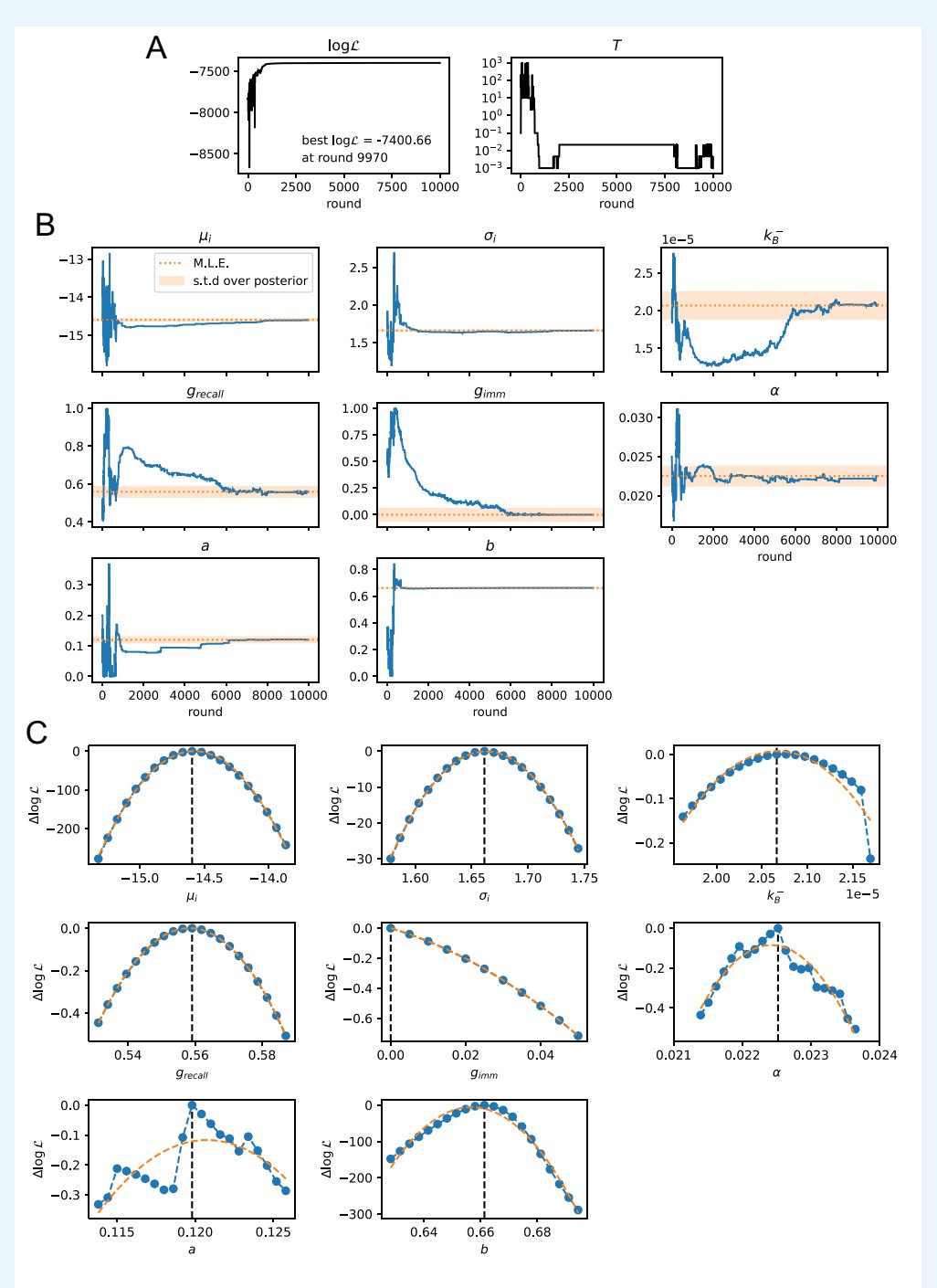

**Appendix 1—figure 6.** Convergence of the stochastic likelihood maximization procedure for variant C of the model (see main text). In this variant 8 of the model parameters are inferred ($\mu_{\mathrm{naive}}$, $\sigma_{\mathrm{naive}}$, $k_B^-$, $\alpha$, $a$, $b$, $g_{\mathrm{recall}}$, $g_{\mathrm{imm}}$). (**A**) Values of the log-likelihood $\log\mathcal{L}$ and the temperature $T$ for the parameter set that reached peak likelihood. (**B**) Evolution of the parameter values during the maximization procedure for the same set (blue lines). Maximum likelihood (ML) estimates of the parameters are marked as orange dashed lines. Orange shaded area cover one standard deviation of the posterior distribution around the ML value, evaluated through a Gaussian approximation of the posterior distribution and the quadratic fit of the log-likelihood displayed in panel C. (**C**) Likelihood variation $\Delta\log\mathcal{L}$ of the posterior distribution for a small ($\pm5\%$) variation of single parameters around their ML values (vertical

black dashed lines). The peaked shapes of $\Delta \log \mathcal{L}$ confirm the convergence of the maximization procedure for all parameters. Orange dashed curves represent quadratic fit of $\Delta \log \mathcal{L}$, used to estimate the orange confidence interval in (**B**).

For completeness in *Appendix 1—figure 7*, we report the comparison between the experimental measurements (orange histograms) and the maximum-likelihood fit of the theoretical model (blue curves) for all the different immunization schemes and all different values of the injected dosage $D$ and time delay between injections $\Delta T$ used for the fit. All these distributions were used for the likelihood evaluation. The normalization of the distributions is done considering only the part inside the experimental sensitivity range. We find good agreement under all different schemes. Moreover we also report the comparison between the theoretical solution and the stochastic simulations of the model (green histograms, average over 1000 stochastic simulations). This comparison shows that for all regimes considered the theoretical solution is a very good approximation to the stochastic model.

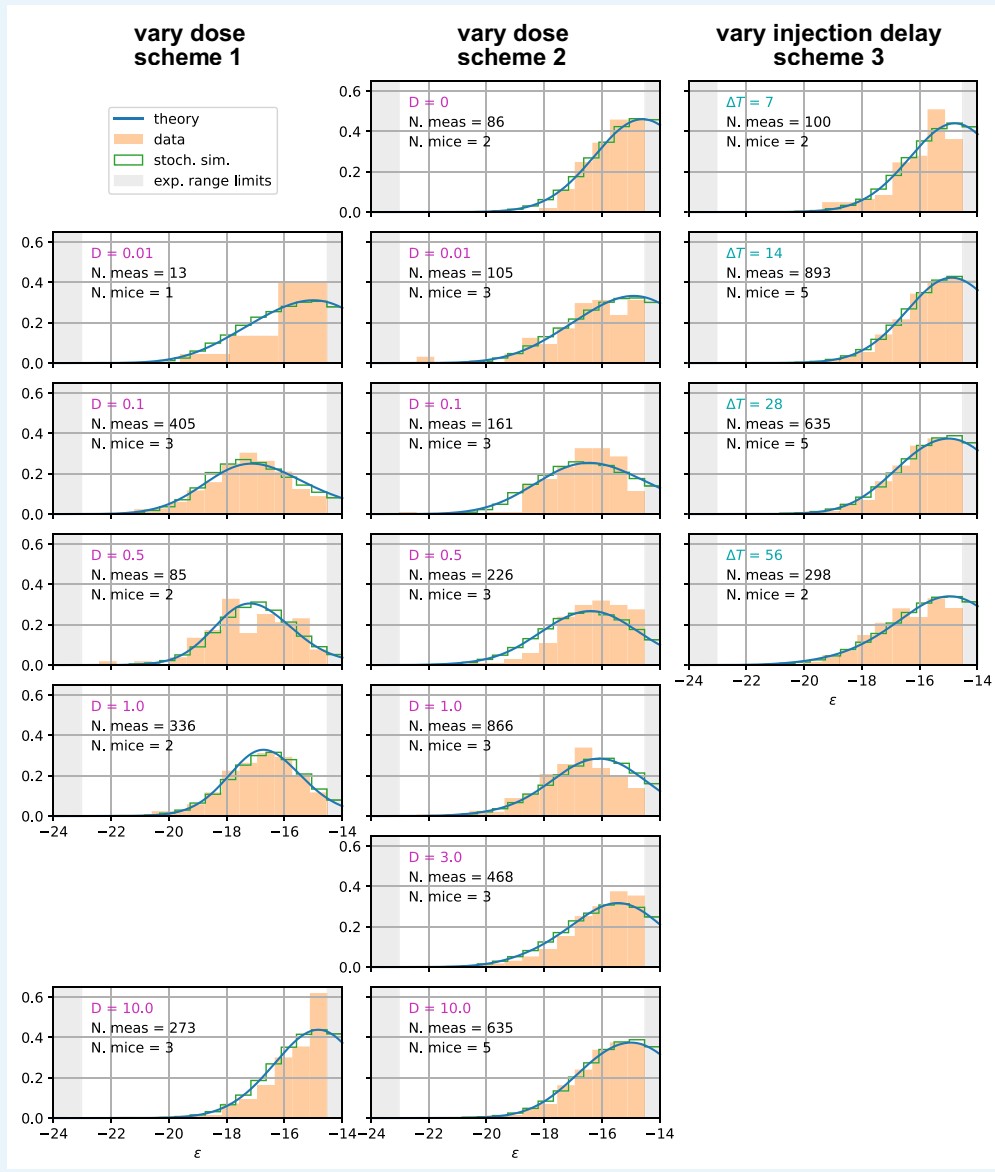

**Appendix 1—figure 7.** Affinity distributions, comparison between experiments and model prediction for all tested conditions. Comparison between experimental measurements

(orange histograms), stochastic model (green histograms), and theoretical solution (blue curve) for affinity distributions of antibody-secreting cells (Ab-SCs) under all the different immunization protocols (see main text for the description of the schemes). Experimental data (orange histograms) consists in measurements of affinities of IgG-secreting cells extracted from mice spleen. The number of mice and single-cell measurements is indicated for each histogram (black). The experimental sensitivity range ($0.1\,\mathrm{nM} \leq K_d \leq 500\,\mathrm{nM}$, or equivalently $-23.03 \geq \epsilon \geq -14.51$) is delimited by the gray shaded area. Blue curves represent the expected binding energy distribution of the Ab-SCs population according to our theory under the same model conditions. For a good comparison all the distributions are normalized so that the area under the curve is unitary for the part inside the experimental sensitivity threshold. For every histogram, we indicate the value of the varied immunization scheme parameter, corresponding to dosage $D$ (pink) for the first two schemes and time delay $\Delta T$ (blue) for the third.

This likelihood maximization procedure is very general and can be easily extended to the inference of any set of parameters in our model, or to other experimental datasets.

## 5. Permissive and stochastic selection: effect of $a$, $b$ parameters

Parameters $a$ and $b$ in **Equations 19, 20** represent respectively the probability for a B-cell to pass or to fail a selection step irrespective from their affinity. Parameter $a$ can be related to the 'permissiveness' of selection, quantifying how likely is for a cell to be positively selected even if its affinity is small. Parameter $b$ conversely encodes the stochasticity in selection, by virtue of which even high affinity cells are not granted survival (e.g. if they don't manage to encounter enough T-cells). For simplicity we define these parameters as constants, but one could immagine that their value may change over time, for example it may be related to the availability of T-cell help. Here we investigate the effect of these parameters on the asymptotic wave-like behavior of the system. This asymptotic behavior is characterized by two quantities: the asymptotic growth rate $\phi(C)$ and the asymptotic maturation speed $u(C)$, as defined in the main text **Equation 3**. These quantities are functions of the Ag concentration $C$. In **Appendix 1—figure 8** we report how these functions change when the parameters $a$ and $b$ are progressively increased. In these tests, we consider only the effect of one parameter at a time and we set the value of the other to its standard value $a = 0.13$, $b = 0.66$. By making the selection more permissive and allowing for survival of even low-affinity cells, parameter $a$ has a double effect on the asymptotic behavior: on the one hand it decreases the asymptotic wave velocity and slows down maturation (**Appendix 1—figure 8E**), and on the other hand it increases the growth rate of the population (**Appendix 1—figure 8C**). On the contrary, increasing parameter $b$ corresponds to increasing the chance that high-affinity cells are selected out of the population. This both decreases the growth rate (**Appendix 1—figure 8D**) and also the maturation speed (**Appendix 1—figure 8F**).

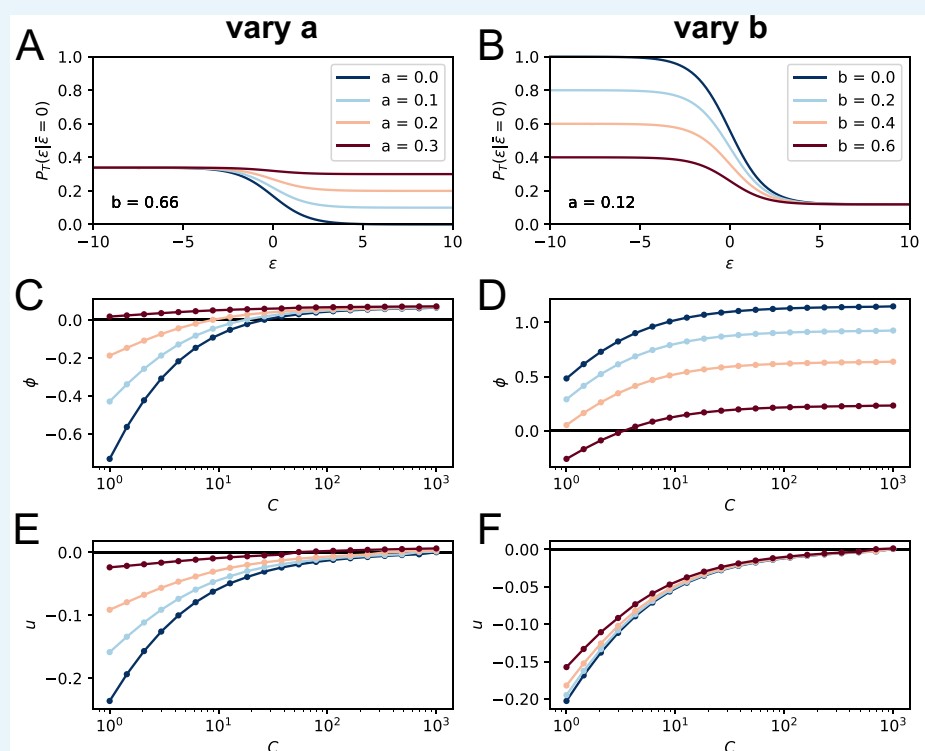

**Appendix 1—figure 8.** Effect of the terms $a$, $b$ on the model asymptotic behavior. On the top row we plot the corresponding T-cell selection survival probability (setting for simplicity $\bar{\epsilon} = 0$ and $C = 1$) respectively in the case $b = 0.66$ and $a$ varying from 0 to 0.3 (A) and $a = 0.12$, $b$ varying from 0 to 0.6 (B). Values of the parameters $a$ and $b$ are color-coded as showed in the legend. In C and E we show the effect of varying $a$ on the asymptotic growth rate $\phi$ and evolution speed $u$ (we set as before $b = 0.66$). Notice how increasing $a$ both slows down evolution and increases the growth rate. In D and F, we report the same quantities for variation of the parameter $b$ (while $a = 0.12$). Increasing $b$ decreases both the growth rate and the maturation speed.

The selection process in affinity maturation has both a *purifying* and *promoting* effect. On the one hand it must negatively select clones that accumulated negative mutations, purifying the population from low-fitness individuals. On the other hand, it must also grant the survival and amplification of the clones that developed affinity-improving mutations. These two properties of selection are weakened by parameters $a$ and $b$ in our model, since they respectively grant survival of low-affinity clones and can cause the removal of high-affinity ones. According to our inference procedure parameters $a$ and $b$ together seem to account for 79% of the selection probability, making so that affinity can make the survival probability vary of only about 21%. This slows down maturation considerably, since it removes any deterrent against accumulating deleterious mutations. The fact that the inference procedure indicated a high value for these parameters suggests that selection in GCs may be permissive, at least for complex Ags, as suggested in *Murugan et al., 2018*.

## 6. Possible model variations

In order to test the relative importance of different model parameters we performed the inference procedure using three different variants for selection (see main text). Variant A corresponds to the inference of all nine inferred model parameters ($\mu_{\text{naive}}$, $\sigma_{\text{naive}}$, $k_B^-$, $\alpha$, $a$, $b$, $g_{\text{recall}}$, $g_{\text{imm}}$, $\epsilon_{\text{Ag}}$). Variant B corresponds to the case in which stochasticity and permissivity parameters $a$ and $b$ are set to zero and only the remaining seven parameters are inferred. In variant C instead Ag-binding selection is neglected, and all eight parameters with the

exclusion of $\epsilon_{\text{Ag}}$ are inferred. The resulting maximum likelihood estimate (MLE) of the parameters, along with the corresponding value of the likelihood, is reported for the three cases in *Appendix 1—figure 9*. The result of the inference procedure show that the removal of Ag-binding selection (variant A vs C, 9 vs 8 parameters) causes only a very modest decrease in log-likelihood, while the removal of stochasticity in T-cell help selection (variant A vs B, 9 vs 7 parameters) generates a consistent log-likelihood decrease. As described in the main text, both the Bayesian Information Criterion (BIC) (*Schwarz, 1978*) and Akaike Information Criterion (AIC) (*Akaike, 1974*) are in support of removing the former but not the latter, and accept variant C.

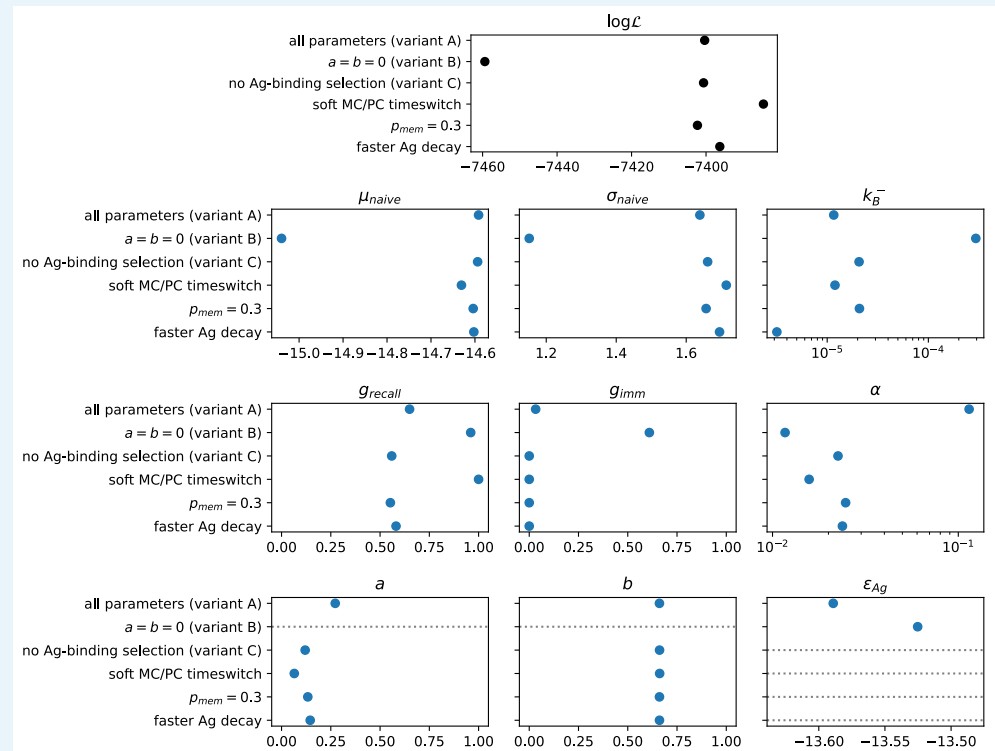

**Appendix 1—figure 9.** Result of the inference procedure on the five variations of the model and comparison with the standard version of the model. The variations are described in appendix sect. 6 (Possible model variations). Top: final maximum value of the log-likelihood obtained with the inference procedure (black). Bottom: maximum-likelihood estimate of all of the inferred parameters (blue). Horizontal grey dashed lines mark parameters that are absent in the variant of the model considered.

Moreover, to show that the model is robust under minor modifications of the hypotheses we consider three minor variations, keeping variant C of selection, and show that they generate similar MLE predictions for the model parameters. First, we test the effect of introducing a residual asymptotic rate of MC/PC production. This case is labelled *soft MC/PC timeswitch*. For simplicity in the model we introduced a complete time-switch between MC and PC production in GCs, making so that asymptotically only PCs are produced for $t \gg \tau_{\text{diff}}$ (cf appendix *Equations 21, 22* and *Appendix 1—figure 1B*). In this modification instead we introduce a residual rate of MC/PC production $\mu_{\text{res}} = 10\%$ modifying appendix *Equations 21, 22* as:

$$\mu_{\text{MC}}(t) = p_{\text{diff}} \left[ \mu_{\text{res}} + (1 - \mu_{\text{res}}) \frac{1}{1 + \exp\{+\frac{t - \tau_{\text{diff}}}{\Delta \tau_{\text{diff}}}\}} \right] \tag{27}$$

$$\mu_{\mathrm{PC}}(t) = p_{\mathrm{diff}} \left[ \mu_{\mathrm{res}} + (1 - \mu_{\mathrm{res}}) \frac{1}{1 + \exp\{ -\frac{t - \tau_{\mathrm{diff}}}{\Delta\tau_{\mathrm{diff}}} \}} \right] \tag{28}$$

This makes so that the fraction of MCs in the differentiated population interpolates between ~90% at small times and ~10% at big times, granting some residual production of MCs at all times. Applying the inference procedure on this more realistic version of the model results in a better final likelihood than the standard (variant C) version. The inferred values of the parameters are on average similar to the ones obtained with the standard version of the model, with the difference of $g_{\mathrm{recall}}$ and $g_{\mathrm{imm}}$. While the inequality $g_{\mathrm{recall}} > g_{\mathrm{imm}}$ still holds, the MLE for these parameter is higher than in the standard case. This can be expected since these parameters control the fraction of MCs in the elicited Ab-SC population, and in this version of the model the MC population contains differentiated cells that would have belonged to the PC compartment in the standard version.

A further modification involves the fraction of seeder clones extracted from the MC population when colonizing a GC. At the second injection some of the seeder clones for the new GC are extracted from the MC population generated following the first injection. The probability of extracting a seeder clone from the MC pool and not from the initial germline distribution depends in the standard version of the model on the number of accumulated memory cells $N_{\mathrm{mem}}$ as $p_{\mathrm{mem}} = N_{\mathrm{mem}}/(N_{\mathrm{mem}} + N_i)$ (cf main text section Stochastic model for affinity maturation). This should account for the fact that intuitively if more MCs were produced in the previous maturation then also more should be recalled. However one could more simply suppose this probability to be constant. We test this case by setting $p_{\mathrm{mem}} = 0.3$. This change generates only a very small likelihood decrease with respect to the standard version of the model, while the MLE for all parameters is almost unchanged.

Finally, we also test the effect of increasing the rate of Ag decay, multiplying it by a factor three (case labelled *faster Ag decay*). This results in a slight increase in the maximum likelihood, and the values of all the model parameters are again compatible with the one of the standard version of the model, with the exception of the Ag consumption rate $k_B^-$ which decreases to compensate the faster decay rate.

# 7. Quantifying beneficial and deleterious mutation events

To quantify the number and impact of mutation events in our simulations we execute 1000 stochastic simulations of a single GC, at an injected Ag dosage of $D = 1\,\mu g$. In each simulation and for each cell we keep track of the number of beneficial and deleterious mutations that each cell accumulates during the course of evolution on the residues we consider ($N_{\mathrm{res}} = 50$, see appendix sect. 1 Model definition and parameters choice). Results are reported in **Appendix 1—figure 10**. In the top row we display the average number of cells for any value of beneficial and deleterious mutations number and at different times: 10, 30 and 50 days after Ag injection. To have an idea of the population size and maturation state at these timepoints one can refer to **Appendix 1—figure 2**, in which stochastic simulations are performed under the same conditions. In our simulations after the first days of maturation deleterious mutations start to appear (see **Appendix 1—figure 10**, t = 10). These are the first mutations to appear since they are much more likely than beneficial mutations (95% vs 5%, see **Appendix 1—figure 1A**). However during the course of evolution these are gradually removed by selection, until eventually beneficial mutations, despite being much rarer, start to dominate (see **Appendix 1—figure 10**, t = 50).

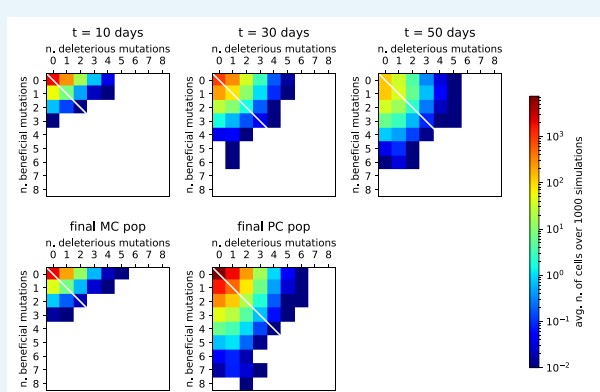

**Appendix 1—figure 10.** Distribution of beneficial and deleterious mutations over 1000 stochastic germinal center simulations at injected Ag dosage $D = 1\,\mu\text{g}$. Color represents the average number of cells that have developed the specified number of beneficial and deleterious mutations for any examined population, according to the color-scale on the right. Top: distribution of mutation numbers in the GC population taken at different times (respectively, 10, 30 or 50 days after Ag injection from left to right). Notice that not all of the populations have the same total size. Bottom: distribution of mutation numbers for the final MC and PC distributions. The average number of total mutations accumulated is 0.13 for MCs and 0.54 for PCs. This number has to be compared to the total number of residues considered ($N_{\text{res}} = 50$).

In the bottom row of **Appendix 1—figure 10** we display in the same way the number of mutations in the MC and PC population. The former is composed of cells that differentiate early ( **Appendix 1—figure 1B** and **Equations 21, 22**) and therefore bear less mutations than the PC population. However, in both cases the vast majority of cells harbor very few mutations, with the average number of mutations per cell being 0.13 for MCs and 0.54 for PCs. The accumulation of more than 4–5 beneficial mutations in a single cell is very rare. These numbers are compatible with experiments performed in a recent work (**Gérard et al., 2020**), in which mice were immunized against Tetanus Toxoid following a protocol similar to the one used in our experiments. The analysis of high-affinity binders showed an average of 6 non-synonymous mutations on the antibody heavy-chain variable region $V_H$ and three mutations in the light-chain variable region $V_L$.

Our model neglects saturation of beneficial mutations, that is the phenomenon by which beneficial mutations cannot be accumulated indefinitely but become rarer as the cell approaches maximum possible affinity. This is partly justified here by the observation that, at least for the inferred value of model parameters, few beneficial mutations are found to accumulate in our simulations. Even when considering clones with the highest number of beneficial mutations, the number of mutations accumulated in our model is compatible with experiments (**Gérard et al., 2020**). As a final remark, notice that even though MCs are not as strongly skewed towards beneficial mutations as PCs, their average affinity is higher than the one of the starting population (cf **Appendix 1—figure 2B**). This is because amongst the founder clones selection will expand the ones having higher affinity (cf **Figure 3**), which will then be overly-represented in the MC population. This shows that in our model maturation is achieved only partially by accumulation of beneficial mutations, the rest being obtained through selective expansion of high-affinity precursors, as also shown in section 'Maturation induces progressive loss of clonality' of the main paper, and confirmed by the strong dependence of the maturation outcome on the initial founder clones population (see **Appendix 1—figure 3E to H**).

# 8. Validation of inference procedure on artificially generated data

To test the reliability of our inference algorithm we generate 10 artificial datasets using our stochastic model and we apply to them the inference procedure. We then compare the inferred values of the parameters with the real value used to generate the data. To verify that the experimental measurements at our disposal are sufficient to infer the model parameters, when generating the data we took into account the size and composition of the experimental dataset. In particular, we generate each artificial dataset as follows. For every experimentally tested conditions (15 conditions in total, consisting in five tested dosages for scheme 1, seven dosages for scheme 2 and 4 delays for scheme 3, with condition $D = 10\,\mu g$ and $\Delta T = 4$ weeks shared between scheme 2 and 3) we run as many stochastic simulations as mice tested for that particular condition. Then from every simulation we extract from the responders population a number of cells equal to the one obtained from each mouse. Extraction of cells is done keeping into account the experimental detection range, therefore we exclude cells having affinity lower that $K_d = 500\,\mathrm{nM}$, and set any affinity higher than the high-affinity threshold $K_d = 0.1\,\mathrm{nM}$ equal to the threshold. Stochastic simulations of each scheme are done in model scenario C, using the standard value of the parameters with one exception: to account for the fact that each mouse contains multiple GCs we raised the number of founder clones in each GC to 2500 instead of 100. This amounts to introducing a greater diversity in the initial population, which in turns reduces the stochasticity in the evolution outcome (see also *Appendix 1—figure 3A to D* and appendix sect. 2 'Stochastic model analysis') and is similar in spirit to averaging between multiple GCs, as it is the case for the experimental measurements of cells extracted from the spleen of mice. This generation procedure was re-executed 10 times, resulting in 10 different datasets. As an example, in *Appendix 1—figure 11B* we show the binding energy distribution of the 10 generated datasets for condition $D = 0.5\,\mu g$ of Ag in scheme 2 (histograms in gray). The histograms are close to the prediction of the deterministic model (blue curves), but with some deviations due to stochastic sampling. The average binding energy of the population for all considered condition is reported in *Appendix 1—figure 11A*. Again, the average binding energy of the generated populations (gray crosses) is close to the prediction of the deterministic model (blue dot), and as expected the noise is higher for the conditions where a smaller amount of experimental measurement was performed (compare with number of measurements displayed in *Appendix 1—figure 7*).

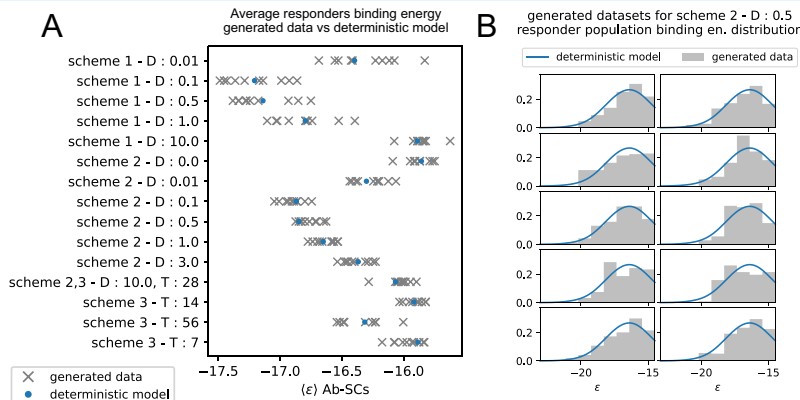

**Appendix 1—figure 11.** Generation of realistic datasets, matching the statistics of the experimental one, for inference procedure validation. (**A**) Average binding energy of responders populations in the 10 artificially generated datasets (gray) for the 15 experimentally tested conditions, vs the same quantity as predicted by simulations of the deterministic model. The condition to which the measurement are referred is reported on the y-axis, in the form of the scheme used, the Ag dosage injected (*D*) and/or the time delay between injection (*T*). (**B**) Distribution of generated binding energies in the 10 generated dataset (gray), compared to

the distribution of binding energies predicted by the deterministic model (blue), for the condition corresponding to the injection of $D = 0.5\,\mu g$ of Ag in scheme 2.

We then run the inference procedure on each artificially generated dataset, using the same setup and initial condition as the ones used to infer model parameters on the real data, under scenario C. The average results of the inference are reported in *Appendix 1—table 1*. For every model parameter, the average inferred value is close to the real value used to generate the data, demonstrating that the amount of experimental measurements at our disposal are sufficient for a good recovery of the model parameters.

**Appendix 1—table 1.** Average results of the inference procedure on the 10 artificially generated datasets. For each model parameter, we report the value used to generate the data (left), and the mean (middle) and standard deviation (right) over the 10 inferred values.

| Parameter | Value used to generate data | Mean of inferred values | Std of inferred values |
|---|---|---|---|
| $\mu_i$ | −14.59 | −14.76 | 0.22 |
| $\sigma_i$ | 1.66 | 1.59 | 0.11 |
| $k_B^-$ | 2.07e-05 | 1.82e-05 | 0.71e-05 |
| $g_{\mathrm{recall}}$ | 0.56 | 0.55 | 0.11 |
| $g_{\mathrm{imm}}$ | 0 | 0.009 | 0.022 |
| $\alpha$ | 0.023 | 0.032 | 0.019 |
| $a$ | 0.120 | 0.125 | 0.094 |
| $b$ | 0.661 | 0.659 | 0.008 |

