## [Decision Letter]

**Acceptance summary:**

This manuscript presents a mathematical model of affinity maturation, which quantitatively fits single cell measurements of B-cell affinities during maturation. The main experimental finding is that maturation speed has an optimum at intermediate antigen dosage. This phenomenon is elegantly explained in the model by a trade-off between selection strength, which is stronger at small antigen dosage, and length of maturation, which is prolonged at high dosage. Overall, this work introduces a novel theoretical framework for B-cell maturation and brings an intuitive insight into the underlying forces that drive the affinity maturation of B-cells.

**Decision letter after peer review:**

Thank you for submitting your article "Quantitative modeling of the effect of antigen dosage on B-cell affinity distributions in maturating germinal centers" for consideration by *eLife*. Your article has been reviewed by Naama Barkai as the Senior Editor, a Reviewing Editor, and two reviewers. The following individuals involved in review of your submission have agreed to reveal their identity: Andreas Mayer (Reviewer #1); William S DeWitt III (Reviewer #3).

The reviewers have discussed the reviews with one another and the Reviewing Editor has drafted this decision to help you prepare a revised submission.

This manuscript presents a mathematical model of affinity maturation, which quantitatively fits single cell measurements of B cell affinity dynamics. The main experimental finding is that maturation speed has an optimum at intermediate antigen dosage. This phenomenon is elegantly explained in the model by a trade-off between selection strength, which is stronger at small antigen dosage, and length of maturation, which is prolonged at high dosage. More broadly, a notable contribution of the work is to introduce a novel theoretical framework, which provides intuitive insight into what drives maturation dynamics and allows more rigorous parameter inference than in prior work.

The reviewers agree that the manuscript presents an interesting approach to address B-cell maturation, but they raise major concerns that we would like to see addressed.

Essential revisions:

1) One major concern is about the compatibility of the proposed dynamics with a number of recent experiments that show a rapid emergence of clonal dominance in a significant fraction of germinal centers (e.g. Tas et al., 2016, Abbott et al., 2018). This incompatibility is certainly concerning. However, we also agree that the data presented in the paper is not a product of a single germinal center (GC) but rather are cells harvested from spleens, which can contain tens of GCs at a given time point. In other words, despite the current claim, which needs to be revised, the manuscript is presenting an effective model for multiple GCs in one spleen. Each of these GCs may contain homogenized populations of cells (consistent with Tas et al.,), but the spleen data may show maintenance of diversity arising from distinct clonal populations established in different GCs.

Given this discrepancy, there are multiple issues that need to be addressed:

1.1) Simulation and modeling extension should be added to systematically include heterogeneous GC structures within a spleen.

1.2) The picture presented in the current work in which the stochastic dynamics within a germinal center is well-described by a deterministic traveling wave dynamics in affinity space seems to be in contradiction with the earlier findings (Tas et al., 2016). Importantly, as evolutionary trajectories approach affinity fitness peaks, it will become increasingly difficult for affinity-increasing mutations to occur. Subsection 'Model limitations and discussion' states "we believe this saturation effect is not relevant to model the limited maturation observed in experiments". This needs more justification, considering the convergent outcomes found in some studies (e.g. Tas et al.,). It's not obvious that the traveling wave asymptote is obtained in GC dynamics.

We propose that the authors explicitly demonstrate a specific scenario (e.g. through simulations) that their model provides at least a correct effective description for evolution of cells harvested from a spleen. Traveling wave approach might be a more realistic effective model for multiple GCs than for affinity maturation in a single GC, as currently presented.

2) The lack of any validation for ML inference procedure is a serious limitation. The manuscript describes a rugged likelihood surface for which convex optimization would be inadequate for arriving at the maximum likelihood estimation (MLE). The authors use parallel tempering to cope with this, which allows sampling across multiple local minima. However, there is no attempt to validate that model parameters can be accurately recovered by this procedure. It is not enough to say that the data are fit well as many points in the high dimensional parameter space may fit the data well. Therefore, the authors should demonstrate that model parameters can reliably be recovered from simulated data for a range of parameter values. It would be more convincing if the authors can further show that recovery is not severely impacted by model assumptions that were adopted for analytical tractability but can be violated in simulations (i.e. model misspecification). Without such validation on simulated data, it is difficult to reliably trust the parameter inference from experimental data.

3) The manuscript describes a computational method for simulation and for inferring parameters from affinity data, but the computational implementation is not made available. Access to the implementation is needed for several reasons: (1) more complete peer review will be possible if reviewers are able to assess the implementation details and even run the code, (2) reproducibility of the results, (3) upon publication, it will be more feasible for other researchers to use or build on this work. We ask the authors to make their code available to the reviewers for the next revision.

---

## [Author Response]

Essential revisions:1) One major concern is about the compatibility of the proposed dynamics with a number of recent experiments that show a rapid emergence of clonal dominance in a significant fraction of germinal centers (e.g. Tas et al., 2016, Abbott et al., 2018). This incompatibility is certainly concerning. However, we also agree that the data presented in the paper is not a product of a single germinal center (GC) but rather are cells harvested from spleens, which can contain tens of GCs at a given time point. In other words, despite the current claim, which needs to be revised, the manuscript is presenting an effective model for multiple GCs in one spleen. Each of these GCs may contain homogenized populations of cells (consistent with Tas et al.,), but the spleen data may show maintenance of diversity arising from distinct clonal populations established in different GCs.Given this discrepancy, there are multiple issues that need to be addressed:1.1) Simulation and modeling extension should be added to systematically include heterogeneous GC structures within a spleen.1.2) The picture presented in the current work in which the stochastic dynamics within a germinal center is well-described by a deterministic traveling wave dynamics in affinity space seems to be in contradiction with the earlier findings (Tas et al., 2016). Importantly, as evolutionary trajectories approach affinity fitness peaks, it will become increasingly difficult for affinity-increasing mutations to occur. Subsection 'Model limitations and discussion' states "we believe this saturation effect is not relevant to model the limited maturation observed in experiments". This needs more justification, considering the convergent outcomes found in some studies (e.g. Tas et al.,). It's not obvious that the traveling wave asymptote is obtained in GC dynamics.We propose that the authors explicitly demonstrate a specific scenario (e.g. through simulations) that their model provides at least a correct effective description for evolution of cells harvested from a spleen. Traveling wave approach might be a more realistic effective model for multiple GCs than for affinity maturation in a single GC, as currently presented.

The reviewers are concerned by the incompatibility of our stochastic evolution model, which gives rise (on average and at 'long' times) to a travelling maturation wave, with recent experimental evidence that GCs can be quickly dominated by single clones. This is an important concern, and we would like to answer comment 1.1 in four points.

First, we fully agree with the reviewers on the relevance of the experiments by Tas et al., (2016) and Abbott et al., (2018), showing various degrees of homogenizing selection (some GCs are quickly dominated by a single clone others stay heterogeneous throughout evolution). To demonstrate that our model can account for this phenomenon we have introduced a minimal modification, consisting in seeding the GC with 100 founder clones (rather than 2,500 as in the previous manuscript). This number was chosen based on Tas et al., (2016)’s estimate ranging from 50 to 200 clones in early GCs. The binding energy of each founder clone is independently extracted from the Gaussian distribution of 'naive' binding energies. During the time of GC formation these clones expand uniformly without mutating, up to a total population of 2,500 cells at day 6 after injection, at which point the GC simulation starts. This modification reduces the diversity of the initial population (but not its size) to a more realistic level. It also allows us to follow the evolution of different clonal families, which was not done in the first version of the manuscript. We observe that this evolution is compatible with the experimental evidence cited above. In particular, in Figure 3 we demonstrate that stochastic simulations of our GCs show different degrees of homogenizing selection (Figure 3A, B), with half of the simulation featuring a clonal family that, by day 28, expands to and 30% of the total GC population (Figure 3C). Moreover, we show that homogenization correlates with the presence of a high-affinity founder (Figure 3D).

Second, considering that GC is seeded by 'only' 100 founder clones rather than 2500 distinct clones drawn from the same naive distribution, as was done in the previous manuscript, has no impact on the average evolution of GCs, but it increases the level of stochasticity of the outcome of the maturation process (cf Appendix 1—Figure 3A to D). Therefore, our deterministic evolution model remains valid and correctly describes the average evolution of one GC.

Third, as rightly pointed out by the reviewers, the cells we measure from spleens are the cumulative outcome of many different GC reactions. While the physical and biological defining constants may vary from GC to GC, this structural variability cannot be inferred from our experimental data that give access to the population averaged affinity distribution. We thus omit this variability and assume that all GCs are defined with the same effective model parameters, but are seeded with different, independently drawn founder clones. This population of heterogeneous GC mature independently from each other, and we collect PCs and MCs at the end of the process. Considering multiple GCs rather than a single one does not have any impact on the average distribution of affinities, which is again given by our deterministic evolution model. Therefore, it is licit to use the average distribution in our inference procedure to extract the effective model parameters from the set of experimentally measured affinities, as explain in the Materials and methods section. Notice that the level of noise (fluctuations around this average distribution) is reduced when averaging over multiple GCs; We take this effect into account through the rescaled level of noise corresponding to the average over 20 different stochastic simulations (see Figure 6).

Fourth, we also agree with the reviewers that the 'traveling wave' description we employ in our paper is exact for the deterministic evolution model only. In other words, the traveling wave is true only on average over the stochastic evolution of one or more GCs.

As a summary, in the amended manuscript, we have changed the initial population of our GC to make it more realistic. Consequently, our stochastic maturation model does show homogenizing selection and the resulting affinity distribution is described on average by a traveling wave. We have made these two essential points much clearer in the new manuscript. We would also like to thank the reviewers for their important comment. We believe that the changes that were introduced following their recommendation, in particular the addition of Figure 3 and of the corresponding paragraph, substantially improved our paper.

We now discuss comment 1.2 and potential saturation effect in mutations. We have quantified the numbers of mutations accumulated by MCs and PCs during the course of evolution in our stochastic model (see Appendix 1—Figure 10 and Appendix 1 subsection 'Quantifying beneficial and deleterious mutation events'). For both populations these numbers are small on average (*<* 1, Appendix 1 subsection 'Quantifying beneficial and deleterious mutation events'), with very few cells accumulating more than 4-5 beneficial mutations. This is especially true for the MC population, in which most of the mutations accumulated are deleterious. This observation shows that maturation in our model mainly comes from simple selection of high-affinity precursor, and only partially by additional mutations increasing their affinity. Consequently, we believe that neglecting the saturation effect of beneficial mutations is an acceptable approximation, at least in the regime of the inferred values of the parameters. In addition, the maximum number of accumulated beneficial mutations (4-5) is also compatible with recent experiments [G´erard et al., 2020], in which mice were immunized against Tetanos Toxoid with a protocol similar to the ones we consider. The authors quantified the mutational load on high-affinity binders (*K_d_*∼ 1 nM) and found an average of 9 non-synonymous mutations on the antibody heavy and light chain variable regions. These experimental results confirm that the number of mutation events considered in our model is not unrealistically high despite the absence of saturation mechanism.

2) The lack of any validation for ML inference procedure is a serious limitation. The manuscript describes a rugged likelihood surface for which convex optimization would be inadequate for arriving at the maximum likelihood estimation (MLE). The authors use parallel tempering to cope with this, which allows sampling across multiple local minima. However, there is no attempt to validate that model parameters can be accurately recovered by this procedure. It is not enough to say that the data are fit well as many points in the high dimensional parameter space may fit the data well. Therefore, the authors should demonstrate that model parameters can reliably be recovered from simulated data for a range of parameter values. It would be more convincing if the authors can further show that recovery is not severely impacted by model assumptions that were adopted for analytical tractability but can be violated in simulations (i.e. model misspecification). Without such validation on simulated data, it is difficult to reliably trust the parameter inference from experimental data.

We certainly agree that our likelihood is a complicated function of the parameters to be inferred, and there is no guarantee that the absolute maximum is found, despite the use of parallel tempering and the care brought to the numerics. Following the reviewers’ advice:

We generated multiple synthetic datasets (cf Appendix 1—Figure 11 and Appendix 1 subsection. 'Validation of inference procedure on artificially generated data'). Each dataset was generated from our stochastic maturation model with the set of parameters inferred from the experimental data and reported in the manuscript. The simulations were carried out under the same immunization schemes as in the experimental dataset, and we extracted the same number of (synthetic) cells from the simulated responders as in the mice we have experimentally studied. In this way, the size and format of each synthetic dataset was identical to the experimental one. Notice that, to gain time, we did not introduce 100 founder clones per GC and averaged over multiple GCs, but rather considered 2500 founder clones; this does not affect the mean affinity distribution and the amplitude of the fluctuations is comparable to what is expected from the average over tens of GC, see Figure 6 and Appendix 1— Figure 3A to D.

The inference procedure was then run on each synthetic dataset. In Appendix 1—table 1 we report the average and standard deviation of the values of the inferred parameters. We find a good match between the results of the multiple inferences and the ground truth values of the parameters (that is, their values inferred from the experimental data and used to generate the synthetic data).

These results, reported in the appendix of the new manuscript, show that our inference procedure is consistent and reliable, despite the roughness of the likelihood and the limited number of available data.

3) The manuscript describes a computational method for simulation and for inferring parameters from affinity data, but the computational implementation is not made available. Access to the implementation is needed for several reasons: (1) more complete peer review will be possible if reviewers are able to assess the implementation details and even run the code, (2) reproducibility of the results, (3) upon publication, it will be more feasible for other researchers to use or build on this work. We ask the authors to make their code available to the reviewers for the next revision.

All the code necessary to produce the figures in the main paper will be made available in the following online repository: https://github.com/mmolari/affinity_maturation. This includes the library to simulate immunization schemes, with both the stochastic and deterministic model, and the inference algorithm. The repository contains a file (parallel tempering.py) that executes the maximum-likelihood inference procedure and recovers the values of parameters reported in the paper. The code that reproduces the figures of the main paper is included in the form of Jupyter notebooks. We also added a short tutorial on how to implement custom immunization scheme simulations, import new datasets and run the inference procedure on them (see readme and docs folder). To maximize readability the code is extensively commented.